# A transcriptomic hourglass in brown algae

Jaruwatana Sodai Lotharukpong[1], Min Zheng[1], Rémy Luthringer[1], Daniel Liesner[1], Hajk-Georg Drost[2,3 ✉] & Susana M. Coelho[1 ✉]

Complex multicellularity has emerged independently across a few eukaryotic lineages and is often associated with the rise of elaborate, tightly coordinated developmental processes[1,2]. How multicellularity and development are interconnected in evolution is a major question in biology. The hourglass model of embryonic evolution depicts how developmental processes are conserved during evolution, and predicts morphological and molecular divergence in early and late embryogenesis, bridged by a conserved mid-embryonic (phylotypic) period linked to the formation of the basic body plan[3,4]. Initially found in animal embryos[5–8], molecular hourglass patterns have recently been proposed for land plants and fungi[9,10]. However, whether the hourglass pattern is an intrinsic feature of all complex multicellular eukaryotes remains unknown. Here we tested the presence of a molecular hourglass in the brown algae, a eukaryotic lineage that has evolved multicellularity independently from animals, fungi and plants[1,11,12]. By exploring transcriptome evolution patterns of brown algae with distinct morphological complexities, we uncovered an hourglass pattern during embryogenesis in morphologically complex species. Filamentous algae without canonical embryogenesis display transcriptome conservation in multicellular stages of the life cycle, whereas unicellular stages are more rapidly evolving. Our findings suggest that transcriptome conservation in brown algae is associated with cell differentiation stages, but is not necessarily linked to embryogenesis. Together with previous work in animals, plants and fungi, we provide further evidence for the generality of a developmental hourglass pattern across complex multicellular eukaryotes.

Multicellularity has evolved multiple times in eukaryotes[13]. This evolutionary transition often resulted in relatively simple life forms, but in some lineages this transition was followed by a series of evolutionary innovations, resulting in more 'complex multicellular' organisms with distinct cell and tissue types, intercellular communication and an intricate developmental programme[1,2]. The emergence of complex multicellularity is thus thought to be a rare event, having occurred independently in animals, fungi, plants, red algae, and brown algae[1]. With the rise of complex multicellular lineages, a major question is how developmental processes accommodate evolutionary change.

A recurring pattern of evolutionary conservation and variation across developmental stages during the life cycle of multicellular organisms was already observed by nineteenth century comparative embryologists, who noticed the marked morphological similarity between embryos[14–17]. This observation was more recently revisited at the molecular level, where the morphological pattern of evolutionary conservation and variation is supported by an analogous pattern at the transcriptomic level[18–20]. Two models have been proposed to describe how conserved developmental processes accommodate evolutionary change: the early conservation model and the developmental hourglass model (as well as hybrids of the two models). According to the early conservation model, evolutionary change is

increasingly permitted as embryogenesis proceeds, which presents a low–mid–high pattern of evolutionary novelty and originates from von Baer's third law of embryogenesis[14]. By contrast, the developmental hourglass model proposes that evolutionary change is restricted in the mid-phase of embryogenesis, presenting a high–low–high pattern of evolutionary novelty. This model is motivated by morphological differences observed in the early phases of embryogenesis (for example, diversity in embryo cleavage patterns), similarity in mid-embryogenesis (as embryos converge on a basic body plan) and differences in the later phases (as embryos acquire further species-specific features)[3,4].

Using transcriptome novelty as a quantitative readout for evolutionary novelty, some early studies have reported early conservation patterns[21–23], whereas more recent studies have reported hourglass patterns across various multicellular eukaryotic lineages using bulk[5–10] and single-cell transcriptomics[24–27]. It should be noted that different biological properties such as pleiotropically expressed genes, mutational robustness, inter-embryo expression variability, chromatin accessibility and enhancer conservation may follow different models[28]. The empirical findings at the transcriptome level further tie in with theoretical modelling, which supports the narrative of the natural emergence of hourglass-like structures in complex evolving systems[29–32].

[1]Department of Algal Development and Evolution, Max Planck Institute for Biology Tübingen, Tübingen, Germany. [2]Computational Biology Group, Department of Molecular Biology, Max Planck Institute for Biology Tübingen, Tübingen, Germany. [3]Digital Biology Group, Division of Computational Biology, School of Life Sciences, University of Dundee, Dundee, UK. ✉e-mail: hdrost001@dundee.ac.uk; susana.coelho@tuebingen.mpg.de

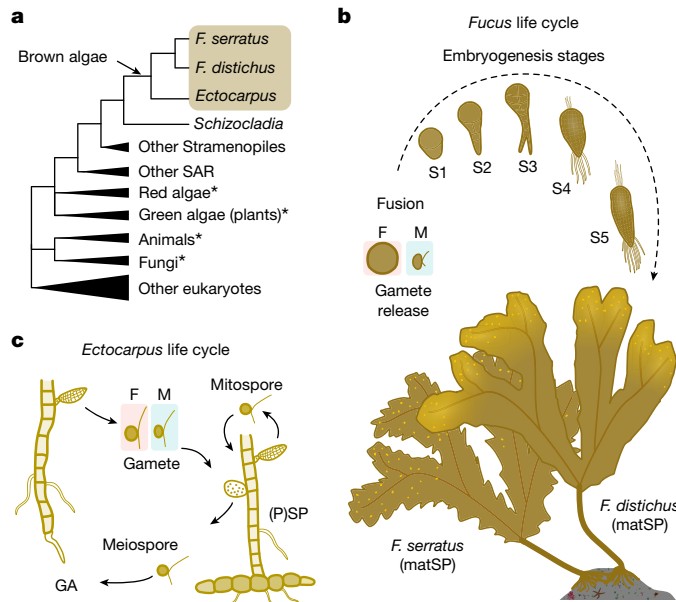

**Fig. 1 | Developmental and morphological diversity in brown algae.**
**a**, Phylogenetic position of *F. serratus*, *F. distichus* and *Ectocarpus* in a simplified eukaryotic tree of life. An arrow marks the independent origin of complex multicellularity in brown algae. The asterisks mark other lineages that evolved complex multicellularity. **b**, The life cycle of *Fucus* species. matSP denotes the sexually mature sporophyte (2n). F, female; m, male. **c**, The life cycle of *Ectocarpus*. Unicellular stages are highlighted in grey. GA denotes multicellular gametophytes (n) and (P)SP denotes both multicellular sporophytes (2n) and morphologically identical partheno-sporophytes (n).

However, our understanding of the generality of the developmental hourglass across complex multicellular eukaryotes is incomplete without considering the brown algae. Brown algae (also called brown seaweeds) belong to the stramenopiles, a large supergroup of organisms that are only distantly related to animals, land plants and fungi[33]. Notably, brown algae independently evolved complex multicellularity 450 million years ago[11,12,34], and have since become the third most morphologically complex lineage on earth, comparable to plants in terms of size, number of cell and tissue types and developmental complexity[33]. In addition, brown algal species vary in relative morphological complexity. For example, the 'morphologically simple' filamentous *Ectocarpus* is composed of up to eight cell types[35], and is capable of multicellular growth and differentiation without canonical embryogenesis[36]. By contrast, the 'morphologically complex' kelps (in the broad sense, including Laminariales and members of Tilopteridales) and Fucales undergo obligatory, canonical embryogenesis to generate metres-long adult individuals composed of dozens of cell types[37]. By harnessing the diversity of morphological complexity in brown algae, we can disentangle the effect of multicellular development per se versus embryogenesis on transcriptome evolution patterns. Furthermore, sexual systems also evolved independently in brown algae[38]. During the course of sexual development in brown algae, most species (for example, *Ectocarpus*) alternate between haploid (gametophyte) and diploid (sporophyte) generations, each consisting of morphologically distinct, multicellular forms connected by three different unicellular stages: gametes, meiospores and mitospores[35] (Fig. 1). We can thus distinguish the potential role of selection in gametes (for example, due to sperm competition) from unicellularity (that is, bottlenecks during the life cycle)[39]. Alongside the convergent evolution of complex multicellularity, these lineage-specific features make brown algae a unique and powerful system to distinguish overlapping processes seen in animal and plant development.

Leaning on this unique natural history, we explore brown algal species exhibiting distinct levels of morphological complexity, to investigate the existence of a developmental hourglass pattern in this group of complex multicellular eukaryotes. If a molecular hourglass pattern does shape development in brown algae, a question arises as to whether the same hourglass model underlies the development of all complex multicellular life.

Here, we propose an experimental design to test whether a molecular hourglass pattern is underlying brown algal development. To approach this, we quantified gene expression levels across key ontogenetic stages for three distinct brown algal species with external development: *Fucus serratus*, *Fucus distichus* and *Ectocarpus* sp. We selected these species to cover the broad diversity of morphological complexity in this group of eukaryotes and because they develop without contamination from parental tissues (Fig. 1a). The Fucales are morphologically complex seaweeds with a well-described embryogenesis that occurs highly synchronously[40]. *F. serratus* has separate male and female sexes, whereas *F. distichus* is a co-sexual species—that is, the same individual produces male and female reproductive structures (Fig. 1b). As a comparative model, we used the filamentous brown alga *Ectocarpus*, which alternates between two simple but morphologically distinct and independently developing forms, the gametophyte and sporophyte. *Ectocarpus* also presents a range of uni- and multicellular stages but not necessarily a canonical embryogenesis[35] (Fig. 1c). To further broaden the species diversity, we analysed transcriptomic data from a subset of developmental stages in two different kelp species, *Laminaria digitata* and *Saccorhiza polyschides*. Both species alternate between morphologically simple gametophytes (which mirrors *Ectocarpus* development) and morphologically complex sporophytes (which mirrors *Fucus* development including embryogenesis).

## Transcriptome evolution in *Fucus* embryogenesis

We used an evolutionary transcriptomics approach[41] to test the developmental hourglass hypothesis for the embryogenesis of the two *Fucus* species. We first collected stage-specific RNA-sequencing (RNA-seq) data (Supplementary Table 1) and assigned phylogenetic ages to each protein-coding gene using GenEra[42] (Extended Data Fig. 1). Combining the expression and evolutionary information, we computed the transcriptome age indices (TAIs) for each stage, which quantifies the weighted mean of the gene age with its transcript expression level[5,41]. In total, we captured the evolutionary and expression data for 8,291 genes in *F. serratus* and 7,907 genes in *F. distichus* (Methods and Supplementary Table 2).

TAI profiles across embryogenesis revealed a transcriptomic hourglass pattern in both *Fucus* species (Fig. 2a,b). The TAI profiles were robust to all RNA-seq data transformations, consistently returning significant *P* values (<0.05) for both the flat line and the reductive hourglass tests[43] (Extended Data Fig. 2 and Methods). We further tested the robustness of the observed hourglass patterns by removing genes with 'noisy' expression profiles using noisyR[44] and confirmed that the resulting hourglass patterns in both species remained largely significant (Extended Data Fig. 3). We note a shift in the timing of the developmental stages between the two species, where the transcriptome of the *F. distichus* stage with the lowest TAI (S4.5) was most similar to the *F. serratus* stage with the lowest TAI (S4) (Extended Data Fig. 4). Extended Data Fig. 4 also highlights transcriptome divergence in early and late stages. Together, these results provide strong evidence that analogous to animals and plants, a developmental hourglass pattern is also shaping embryogenesis of both *Fucus* species.

Of note, we observed that the expression of evolutionarily young genes (that is, genes associated with the origin of complex multicellularity in brown algae in phylostratum (PS) 7, and species-specific genes in PS 8) were markedly lower during the waist of the hourglass, as opposed to the expression of older genes being higher at this stage

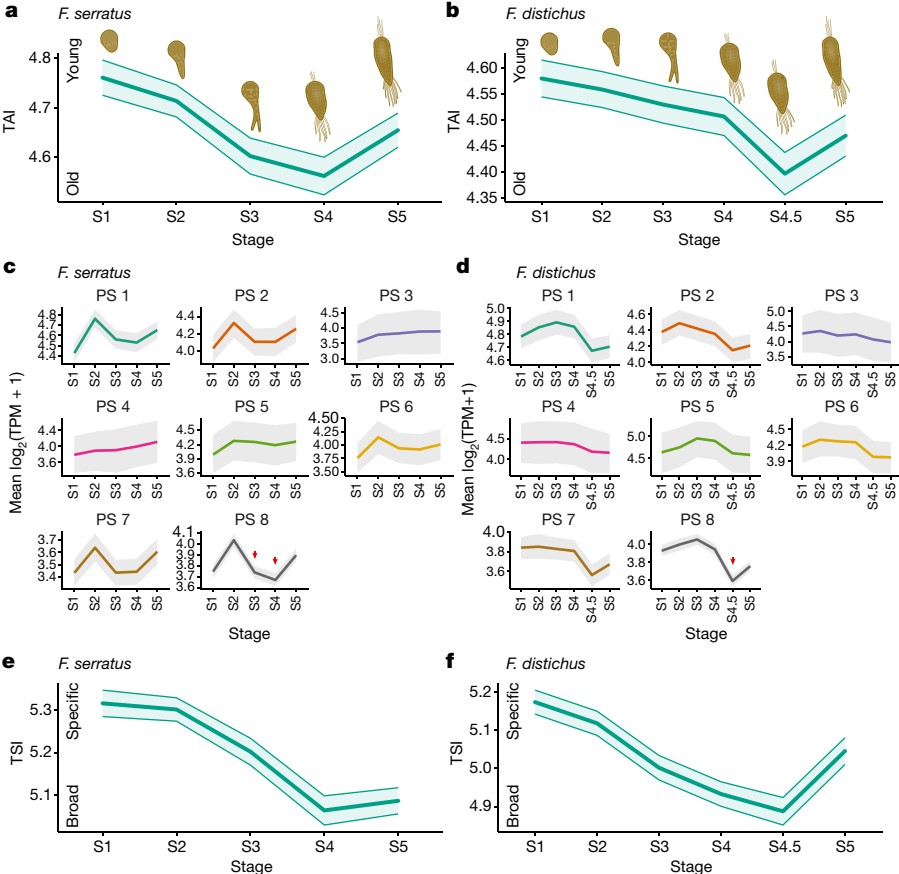

**Fig. 2 | Presence of molecular hourglass patterns in *Fucus* embryogenesis.** **a**,**b**, TAI profile across embryo stages in *F. serratus* (**a**) and *F. distichus* (**b**). A lower TAI marks an older transcriptome (that is, composed of evolutionarily older genes) and vice versa. **c**,**d**, The mean expression profile of each phylostratum (PS) across embryo stages in *F. serratus* (**c**) and *F. distichus* (**d**). Note that expression is restricted in the category of evolutionarily youngest genes (PS 8) in both

species, as indicated by red arrows. TPM, transcripts per million. **e**,**f**, TSI profile of embryogenesis in *F. serratus* (**e**) and *F. distichus* (**f**). A lower TSI marks a transcriptome composed of more broadly expressed (temporally pleiotropic) genes and *vice versa*. 50,000 bootstraps were used to compute the s.d. in **a**,**b**,**e**,**f** and the 95% confidence interval **c**,**d**.

(Fig. 2c,d). Therefore, the repression of expression of young genes may underlie the waist of the hourglass, recapitulating observations in other systems[5,9,45].

It has been proposed that the conserved transcriptome composition in the waist of the hourglass is caused by higher pleiotropy of genes expressed in these stages[46,47]. We examined stage-specific expression using *tau* as an estimate for temporal pleiotropy[48,49], and computed the resulting transcriptome specificity index (TSI) profile across developmental stages. Stages corresponding to the waist of the hourglass (S4 and S4.5 for *F. serratus* and *F. distichus*, respectively) were represented by more broadly expressed genes (that is, low *tau*) whereas early and later developmental stages were characterized by more stage-specific genes (Fig. 2e,f and Supplementary Table 3). Notably, gene expression breadth correlates with the number of protein–protein interactants, developmental essentiality and expression quantitative trait locus-based pleiotropy measures in other systems[46,50]. In sum, more broadly expressed (potentially pleiotropic) genes are associated with the evolutionarily conserved stages of development in both *Fucus* species, consistent with findings in animals[46,47].

Finally, we investigated the possible biological processes that underlie transcriptome evolution patterns during *Fucus* embryogenesis. Of note, the waist of the hourglass in both *Fucus* species paralleled a major ontogenetic transition, from a 'cell-type differentiation' stage, in which the algal body plan is established, to a more 'proliferative' stage, in which development is mainly characterized by somatic cell divisions leading to expansion in the size of the organism (Extended Data Fig. 5).

This finding mirrors similar observations in animals and plants, in which transcriptomic hourglass patterns mark major ontogenetic transitions (for example, cell fate acquisition to differentiated cell growth)[18,45].

Together, our observations demonstrate that both *Fucus* species display a transcriptomic hourglass pattern. The conserved mid-embryonic period is characterized by reduced expression of evolutionarily young genes and a relatively higher expression of more broadly expressed genes, and corresponds with a major developmental transition from cell fate determination to cell proliferation.

## Transcriptome evolution in *Fucus* adults

In animals and plants, the degree of transcriptome conservation also differs in life cycle stages outside embryogenesis (such as between tissues or sexes). For example, evolutionarily young genes are disproportionately expressed in the testis of nematodes, flies and mammals[51–54], and in male reproductive cells in plants[55,56]. To test whether these patterns are present in brown algae, we examined the TAI across different adult tissues of the two *Fucus* species, as well as between sexes (Fig. 3a).

In *F. serratus* males, TAI values differed significantly between tissues (flat line test; $P = 8.57 \times 10^{-4}$), with reproductive tissues (reproductive tip) exhibiting a markedly higher TAI compared to vegetative tissues (vegetative tip, holdfast and stipe) (pairwise TAI test; $P = 0.00534$) (Fig. 3b). This difference was also observed in *F. distichus* (pairwise TAI test; $P = 0.00240$) (Fig. 3c). By contrast, in *F. serratus* females, TAI values did not differ significantly between vegetative and reproductive tissues

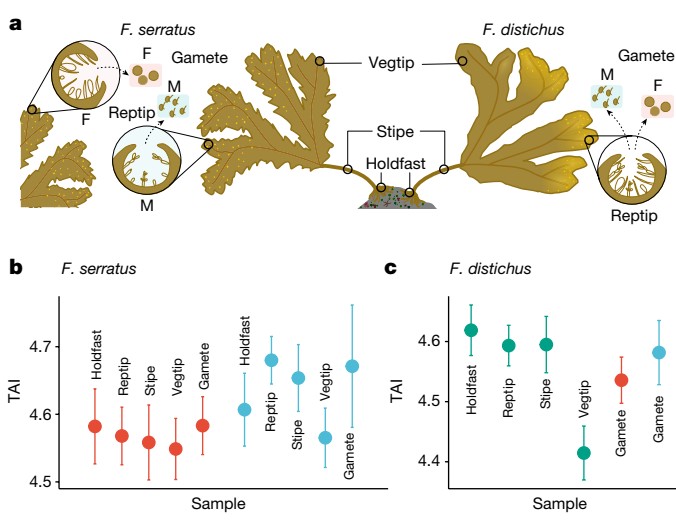

**Fig. 3 | Transcriptome evolution landscape across sex and tissues in adult *Fucus*. a**, Outline of adult (mature sporophyte) tissues and gametes in *Fucus* species. Note, *F. serratus* has separate male and female sexes, whereas in the co-sexual *F. distichus*, the same individual produces both male and female gametes. Reptip, reproductive tip; vegtip, vegetative tip. **b**, TAI profile across adult tissues and gametes in *F. serratus*. **c**, TAI profile across adult tissues and gametes in *F. distichus*. A lower TAI marks a transcriptome composed of evolutionarily older genes and vice versa. We used 50,000 bootstraps to compute the s.d.

(pairwise TAI test; $P = 0.407$) (Fig. 3b). We also noticed that sexual differences of TAI were more pronounced at the reproductive tip in *F. serratus* adults, with males having a markedly younger transcriptome compared to females (pairwise TAI test; $P = 2.59 \times 10^{-6}$). Notably, the stipe (in both species) also presented relatively young transcriptomes, specifically in males of *F. serratus* (Supplementary Table 4). Therefore, similar to animals and plants, the levels of conservation of transcriptomic patterns are variable across the life cycle of *Fucus*, with *Fucus* males (particularly reproductive tissue) displaying an evolutionarily younger transcriptome.

## Transcriptome evolution in *Ectocarpus*

In contrast to *Fucus*, *Ectocarpus* morphology is substantially simpler, with fewer cell and tissue types, developing in absence of canonical embryogenesis[35,36]. Its life cycle comprises two morphologically distinct, free living multicellular generations, the gametophyte and the sporophyte (see Fig. 1c), each composed of three to five cell types[35]. During this alternation of generations, a total of three types of unicellular units are produced: gametes, meiospores and mitospores (see Fig. 1c), which enables us to disentangle the effect of being a 'gamete' per se from the effect of being a unicellular unit. Furthermore, *Ectocarpus* gametes can develop parthenogenically (without gamete fusion) into haploid adults (parthenosporophyte) whose morphology closely resembles that of a diploid sporophyte[36] (see Fig. 1). Thus, *Ectocarpus* is a powerful comparative system to test whether the hourglass pattern seen during *Fucus* embryogenesis is the result of constraints imposed by multicellular development per se or whether this hourglass pattern of transcriptome conservation is tied to a specific embryogenesis process present in *Fucus* but not in *Ectocarpus*.

We examined the developmental transcriptome during the *Ectocarpus* parthenogenetic life cycle by profiling the transcriptomes of the three unicellular stages (gametes, meiospores and mitospores), three stages during gametophyte development (immature, mature and senescent), and three stages in parthenosporophyte development (early,

immature and mature), for both male and female lines (Extended Data Fig. 5 and Supplementary Table 1). As done in *Fucus*, we performed gene age inference (Extended Data Fig. 1 and Methods), and computed the TAI values from 11,571 genes at each developmental stage (Supplementary Table 2). This analysis revealed that *Ectocarpus* unicellular stages in both males and females exhibited a significantly higher TAI compared with the multicellular stages (Fig. 4a and Extended Data Fig. 6), and this pattern was consistent across RNA-seq data transformations (Extended Data Fig. 7). When restricting the reductive hourglass test to multicellular gametophytic or partheno-sporophytic development, only the male partheno-sporophytes returned a significant hourglass shape (reductive hourglass test; $P = 4.22 \times 10^{-15}$). Since this pattern is not consistent in other instances of multicellular development (male gametophyte, female gametophyte and parthenosporophyte) (Supplementary Table 4), we interpret that multicellular stages in these filamentous brown algae have a lower TAI overall, rather than displaying a canonical transcriptomic hourglass. Moreover, conservation in the multicellular stages according to the TAI patterns was recapitulated when computing transcriptome distance/similarity between life cycle stages in *Ectocarpus* (both males and females) and the embryo stages in *Fucus* (Extended Data Fig. 8).

In addition, we computed the average purifying selection acting on each stage through the transcriptome divergence index (TDI) metric, where a lower TDI indicates a transcriptome composed of genes under stronger purifying selection and vice versa (Methods, Supplementary Table 5). Compared with the TAI data, we observed that gametes, but not spores, consistently exhibited a high TDI (Fig. 4b). Our observations suggest that whereas evolutionarily young genes are more likely to be expressed in unicellular stages, genes under relaxed purifying selection are disproportionately found in gametes.

The TAI profile was markedly different between males and females during multicellular development (Fig. 4a and Extended Data Fig. 6). *Ectocarpus* males had higher TAI values than females during the early parthenosporophyte stage, whereas the reverse occurs during the immature gametophyte and parthenosporophyte stages. Of note, the sexual difference in TAI during the gametophyte development culminated at the immature stage, which is the stage when the gametophyte transcriptomes are most different between males and females[57].

We further focused on the genes that underlie the high TAI values in unicellular stages. Based on the partial TAI value of each individual gene, $\mathrm{pTAI}_i = \mathrm{ps}_i \times e_{is} / \sum_{i=1}^{n} e_{is}$ (where $e_{is}$ denotes the expression level of a given gene $i$ at stage $s$, $ps_i$ is its gene age assignment, and $n$ is the total number of genes)[41], we identified 115 genes in males and 98 in females that most strongly contributed to the TAI across all unicellular stages (Extended Data Fig. 9) (Methods). Gene Ontology (GO) term analysis did not retrieve any functional enrichment, especially since fewer than 10% of these genes have functional annotation. By contrast, the same analysis using genes with the strongest contribution to the TAI in the multicellular stages returned older genes with GO term enrichment linked to translational, organellar and transcriptional processes in both males and females (Extended Data Fig. 10 and Supplementary Table 6).

Finally, to broaden the species diversity and thus further test the generality of transcriptome conservation during multicellular development in brown algae, we inferred gene age (Extended Data Fig. 1) and computed TAI in the kelps *L. digitata* and *S. polyschides* (using 16,298 and 15,030 genes, respectively) (Supplementary Table 2 and Methods). Both species alternate between morphologically complex sporophyte and morphologically simple filamentous gametophyte generations (Extended Data Figs. 11 and 12). In the sporophyte development of *L. digitata*, we observed the lowest TAI during an embryo stage, whose transcriptome is closest to the most conserved stages in *Fucus* (Extended Data Fig. 12). Meanwhile, in the kelp gametophyte development, we observed a lower TAI during multicellular stages compared to unicellular samples (Extended Data Fig. 12), similar to *Ectocarpus*.

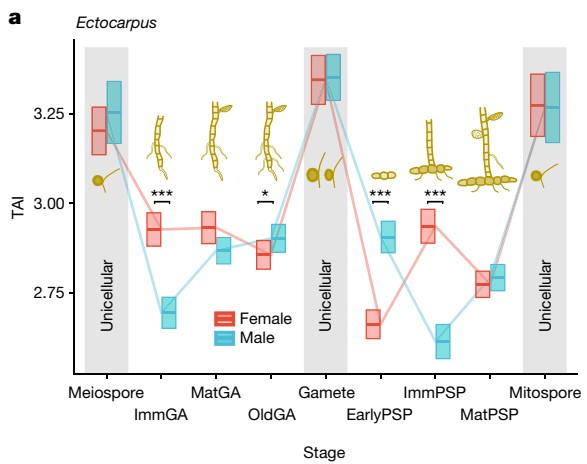

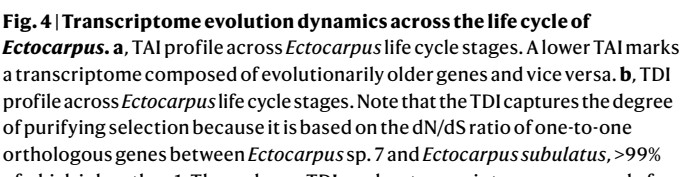

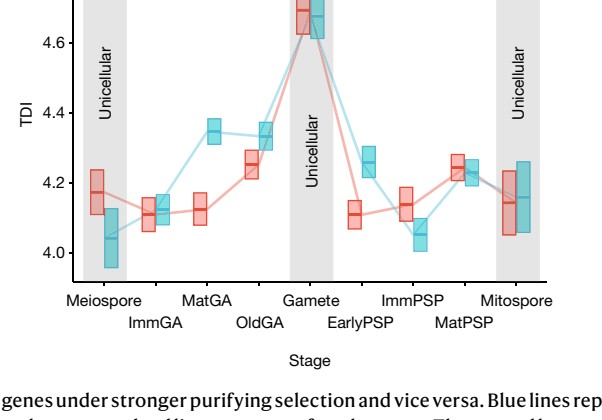

**Fig. 4 | Transcriptome evolution dynamics across the life cycle of Ectocarpus. a**, TAI profile across *Ectocarpus* life cycle stages. A lower TAI marks a transcriptome composed of evolutionarily older genes and vice versa. **b**, TDI profile across *Ectocarpus* life cycle stages. Note that the TDI captures the degree of purifying selection because it is based on the dN/dS ratio of one-to-one orthologous genes between *Ectocarpus* sp. 7 and *Ectocarpus subulatus*, >99% of which is less than 1. Thus, a lower TDI marks a transcriptome composed of genes under stronger purifying selection and vice versa. Blue lines represent male stages and red lines represent female stages. The top and bottom limits of the boxes demarcate the s.d., based on 50,000 bootstraps. Life cycle stages (*x* axis) are denoted as follows: GA, gametophyte; PSP, parthenosporophyte; Early, early (2–5 cell) stages; Imm, sexually immature stages; Mat, sexually mature stages.

In sum, whereas in the morphologically complex species or generations, the period of transcriptome conservation was coupled to the transition from cell differentiation to proliferation during embryogenesis, in the morphologically simple *Ectocarpus* as well as in the gametophytes of *L. digitata* and *S. polyschides*, the switch to multicellularity from unicellular stages constrains transcriptome evolution.

## Discussion

The developmental hourglass model has been reported (and debated) at the molecular level in animals[5–8,21–25,28], plants[9,27,43,45,56,58], and fungi[10,59,60]. Here we examined the prevalence of the transcriptomic hourglass pattern in an independently evolved complex multicellular lineage, the brown algae.

### A transcriptomic hourglass in *Fucus*

We show that embryogenesis of morphologically complex brown algae is underpinned by a transcriptome evolution pattern that is consistent with an hourglass model of embryonic evolution. *Fucus* embryos were more divergent at the earliest and latest stages of embryogenesis but presented a more conserved transcriptome during the mid-embryonic period, which serves as a body plan blueprint for the adult organism.

The hourglass pattern in *Fucus* was associated with reduced expression of evolutionarily young genes, rather than the upregulation of evolutionarily older genes, during the conserved mid-embryonic period, analogous to observations in animals and plants[5,9,24,45]. This pattern, together with data from kelp sporophytes, suggests that ancient genes (rather than genes specific to brown algae) form the *Fucus* and kelp body plans, pointing to a *cis*-regulatory hypothesis for the redeployment of pre-existing genes in the evolutionary innovations associated with brown algal embryogenesis[61]. Moreover, more broadly expressed, potentially pleiotropic, genes were associated with the evolutionarily conserved stages of development in *Fucus*, mirroring animal embryogenesis[46,47]. Notably, the waist of the hourglass corresponded to a major ontogenetic transition, from a cell-type differentiation stage where the body plan of the adult *Fucus* is established, to a proliferative stage, where development is largely characterized by growth of established cell types. This observation is reminiscent of the transition from primitive development to definitive development[62,63] and is consistent with the 'organizational checkpoint' hypothesis, which postulates that a major transcriptome switch during transitions from cell fate acquisition into multicellular growth phases underlies the transcriptomic hourglass seen in animals, plants and fungi[18].

The evidence that we present for the hourglass model, particularly the evolutionary novelty in the early stages, follows a lineage of studies on the establishment of polarity in brown algal zygotes[64], which details variability in early embryogenesis. For example, the role of maternal cytoplasmic contribution and extrinsic cues differ markedly between *Fucus*[65,66], *Dictyota*[67] and *Saccharina*[68]. These observations are consistent with the variations that are also seen in early animal and plant development, which is followed by a more conserved period in mid-embryogenesis[69]. We suspect that the hourglass model also describes development in morphologically complex brown algae beyond the transcriptomic level.

### Young genes in reproductive tissues

Transcriptome profiling of adult *Fucus* tissues revealed that male reproductive organs express a younger transcriptome, suggesting that evolutionarily young genes are more permissively expressed. This pattern is likely associated with the presence of male gametes (sperm) in the reproductive organs of male individuals. These observations complement independent findings in animals and plants where young genes are enriched in male reproductive tissue[51–56]. The younger transcriptome of *Fucus* sperm may be associated with sexual selection, which acts mainly through gamete-level interactions in sessile broadcast spawners such as *Fucus*[70]. Furthermore, the expression of younger genes in *Fucus* males is consistent with recent findings in the brown alga *Macrocystis*, implicating the female development programme as the morphogenetic 'ground state', superimposed upon in males[71,72], though the mechanism behind this pattern in non-reproductive tissues is unclear.

By contrast, *Ectocarpus* did not exhibit differences in transcriptome age in male compared to female gametes. This result is consistent with the low level of sexual dimorphism typical of near-isogamous species[57,73], limiting the extent of sexual selection in males compared to females, as well as the smaller set of sex-biased genes in *Ectocarpus* compared to *Fucus* and *Macrocystis*[74].

### Multicellularity constrains transcriptome evolution

Unlike *Fucus*, *Ectocarpus* development consists of two independent, multicellular, morphologically simple life stages connected by three

types of unicellular stages (gametes, meiospores and mitospores). Crucially, the full life cycle can proceed without a canonical embryogenesis[36]. Although we did not find a classical 'hourglass' signature in *Ectocarpus*, we did find that multicellular developmental stages exhibit more conserved transcriptomes compared to unicellular stages. This finding was also supported in the gametophytes of two kelp species, *L. digitata* and *S. polyschides*. Furthermore, the low TDI associated with multicellular stages of development likely reflects ongoing purifying selection, implicating multicellularity per se as under evolutionary constraint. This evolutionary pattern in filamentous life stages may be due to the lack of a singular 'mid-embryonic period', where the body plan is established. Instead, *Ectocarpus* (as well as *L. digitata* and *S. polyschides* gametophytes) exhibits a 'modular' development, in which cell types are differentiated continuously over time, reiterating a filamentous body plan. We reason that the regulatory complexity from cell-type differentiation programmes activated across filamentous multicellular stages results in an overall conserved transcriptome. We posit that these constraints may have become concentrated towards a more singular 'mid-embryonic period' during the evolution of lineages with increased morphological complexity, such as *Fucus*.

In conjunction with the pattern of 'multicellularity constraint', unicellular dispersal stages may be more permissive to evolutionary change[75]. Differences in the cell structure of unicells, such as the lack of a cell wall, can result in (a)biotic exposure which opens new selection opportunities[76] compared with multicellular stages. For example, it is well known that virus infection occurs in the unicellular stages (gametes and spores) in *Ectocarpus*[77]. Of note, whereas TAI was similarly high for all unicellular stages, only gametes show decreased purifying selection, suggesting a signal for sexual evolution at the sequence substitution level that is specific to gametes. Our results demonstrate that species that lack multicellular organization via embryogenesis may still exhibit a developmental window with higher transcriptome conservation compared to unicellular stages.

Together, we present evidence for the existence of a developmental hourglass pattern during embryogenesis in morphologically complex brown algae, analogous to hourglass patterns previously reported in animals, plants and fungi. Our distinction between complex multicellular development and embryogenesis suggests that transcriptome conservation patterns are a fundamental characteristic of complex multicellularity itself, with possible downstream effects on embryogenesis.

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

## Methods

### Sample preparation

Details of algal strains used are described in Supplementary Table 1. *Fucus* embryos were prepared as described previously[78]. In brief, gametes were allowed to release in 2 l natural seawater (NSW), and cleaned using several NSW baths for female gametes and phototaxy for male gametes. Then, gametes were mixed together for 1 h and fresh zygotes were cleaned as for female gametes. *F. serratus* and *F. distichus* embryos grow highly synchronously and at least 10,000 developing embryos were flash frozen at specific developmental stages (see Extended Data Fig. 5). Embryos were grown at 14 °C in NSW (*F. serratus*) or at 10 °C in diluted NSW (*F. distichus*) supplemented, for one week, with $GeO_2$ (0.4 mg l⁻¹) to avoid growth of diatoms. Media was changed weekly. For both species of *Fucus*, embryos were grown under neutral day conditions (12 h/12 h day/night cycle). *Ectocarpus* life cycle stages were grown at 14 °C in Provasoli-enriched NSW (PES)[79] a 12 h/12 h day/night cycle and 20 µmol photons m⁻² s⁻¹ irradiance, as described previously[80].

Kelp gametophytes were grown vegetatively in 50% PES with iodine enrichment[79,81] at 14 °C under red light in a 14 h/10 h day/night cycle. To induce fertility, gametophyte tufts were carefully ground using a mortar and pestle and the gametophyte fragments were sown at low density (approximately 500 gametophytes per cm²). Gametophyte maturity was induced at 12 °C and 20–30 µmol photons m⁻² s⁻¹ white light under a 16 h/8 h day/night cycle in full-strength PES. Fertile gametophytes with visible gametangia were collected after seven days. Both sexes were co-cultivated to induce maturity and sporophyte production. Field samples were collected from Santec (France), Perharidy (France) and Kiel (Germany) as described in Supplementary Table 1.

Note that *Fucus* and *Ectocarpus* are broadcast spawners, releasing their gametes in the surrounding seawater, where subsequent development takes place. Early development can therefore be followed in a large number of replicate individual clones that develop highly synchronously, without potentially contaminating parental tissue, greatly facilitating experimental approaches. Meanwhile, in *L. digitata* and *S. polyschides*, access to biological material is challenging, resulting in a less complete dataset for these species. For example, obtaining sufficient quantities of very early sporophyte stages of *L. digitata* was impeded by their attachment to the maternal gametophyte.

### RNA extraction from brown algae

Adult tissues of *Fucus* and kelps were quickly brushed and then rinsed with filtered and autoclaved NSW. Different parts such as holdfast, stipe, meristem, vegetative tissue and reproductive tip were sliced into 0.5–1 cm pieces and transferred into 1.5 ml low-bind Eppendorf tubes. The tubes were snap frozen in liquid $N_2$ and the stored at −80 °C until further processing, as done for other stages of *Fucus* and kelp development as well as *Ectocarpus*.

The RNA extraction protocol followed previous publications[71,82] and is described in Supplementary Table 1. Snap-frozen algae were dry-ground with a pestle in liquid $N_2$/dry ice and mixed with 750 µl of freshly prepared RNA-extraction buffer (100 mM Tris-HCl pH8.0 (Thermo Fisher AM9856); 1.4 M NaCl; 2% CTAB (Sigma Aldrich, 52365-50G); 20 mM EDTA pH8.0; 1% β-mercaptoethanol; 2% polyvinylpyrrolidon (Thermo Fisher AM9690)) preheated to 65 °C. Then 250 µl 5 M NaCl were added into the tubes. An equal volume of chloroform: isoamylalcohol (24:1) was added and mixed well followed by centrifugation at 10,000*g* for 15 min at 4 °C. The aqueous phase was removed into RNAse-free tubes and extracted again with 250 µl pure ethanol and an equal volume of chloroform: isoamylalcohol (24:1) as before. RNA was precipitated by adding LiCl (Thermo Fisher, AM9480) to a final concentration of 4 M, together with 1% volume of β-mercaptoethanol, mixing and incubating at −20 °C overnight.

The RNA was pelleted by centrifugation at full speed (>18,000*g*) for 45 min to 1 h at 4 °C. RNA was washed with 70% cold ethanol and the pellet was air dried for 3–5 min and then the RNA was dissolved in 30 µl RNAse-free $H_2O$. Residual DNA was eliminated using the TURBO DNase Kit (Thermo Fisher, AM1907) according to the manufacturer's instructions. The final RNA concentration and size distribution were determined using a Qubit (RNA BR Assay Kit, Invitrogen, Q10210) and an RNA Nano bioanalyzer (Agilent, 5067-1511).

### RNA-seq

The RNA-seq libraries were prepared using commercially available kits according to the manufacturer's instructions. Poly-A selection for mRNA enrichment was performed using the corresponding NEB kit (E7490) followed by library preparation using the directional RNA library prep kit from NEB (E7765). A Single Cell/Low Input RNA library prep kit (NEB, E6420) was used to synthesize additional cDNA and prepare sequencing libraries for samples where it was not possible to obtain large amounts of material (meiospores and mitospores in *Ectocarpus*; earlySP in *L. digitata*; matGA in *S. polyschides*) (Supplementary Table 1). The Qubit 1× dsDNA HS assay kit (Invitrogen, Q33230) was used to determine the final DNA concentration of the libraries and the DNA high-sensitive Kit (Agilent, 5067−4626) was employed for bioanalyzer analysis to evaluate the distribution of insert sizes.

Sequencing was performed on a NextSeq2000 instrument with sequencing kit P3-300 (Illumina). The libraries were pooled for sequencing such that for each library we obtained about 30,000,000 reads, corresponding to 9 Gb of data (Supplementary Table 1).

RNA-seq datasets were processed using the nf-core/rnaseq pipeline v3.5[83,84]. For all three species, expression quantification was performed using salmon v1.5.2[85], to ensure consistency, and imported to R using tximport v1.26.1[86]. For *Ectocarpus*, RNA-seq reads were pseudo-mapped to transcripts inferred for each gene from version 2 of the *Ectocarpus* species 7 genome[11]. For *L. digitata* and *S. polyschides*, the same method was applied to recently published genomes[12]. Previously published *S. polyschides* immGA data[71] was remapped to maintain consistency. Since high quality public genomes were not available for *F. serratus* and *F. distichus*, the quantification was carried out on recently published de novo transcriptome assemblies[74].

We precluded genes with mean length-scaled TPM (transcripts per million) across samples below 2 from subsequent analyses. In analyses indicated as 'denoised', we further removed genes with noise-like behaviour using the 'counts' mode of noisyR v1.0.0[44].

### Transcriptome age index

The TAI captures the average gene age of a given transcriptome, weighted by the expression level of each gene[5,41]. The relative age of each gene in *Ectocarpus*, *F. serratus*, *F. distichus*, *L. digitata* and *S. polyschides* was inferred using GenEra v1.0[42], based on genomic phylostratigraphy[87]. In brief, GenEra takes all protein-coding genes and pairwise aligns these sequences against the taxonomy-resolved NCBI non-redundant database[88,89], using DIAMOND v2.0.14 ('sensitive' mode; e-value < 10⁻⁵)[90]. Next, search hits are filtered by their distribution across taxonomic nodes until the most distant taxonomic node is determined as the 'gene age' (or removed as potential contamination), with the evolutionarily oldest genes assigned as phylostratum (PS) 1 and the youngest assigned as PS 8 in *F. serratus*, *F. distichus* and *S. polyschides*, PS 10 in *L. digitata* and PS 11 in *Ectocarpus*. PS 7 corresponds to the origin of brown algae (complex multicellularity). For genes with more than one isoform, the age of the oldest isoform was used. Thus, after filtering lowly expressed genes across all samples (TPM < 2) and potential contaminations, we obtained expression and evolutionary data for 8,291 genes in *F. serratus*, 7,907 genes in *F. distichus* and 11,571 genes in *Ectocarpus*. Using myTAI v1.0.1.9000[41], TAI was calculated for each stage (TAI$_s$) as follows,

$$TAI_s = \sum_{i=1}^{n} \left( \frac{ps_i \cdot e_{is}}{\sum_{i=1}^{n} e_{is}} \right)$$

where $ps_i$ denotes the relative gene age (phylostratum) for a given gene $i$. The term $e_{is}$ denotes the expression level of a given gene $i$ at developmental stage $s$ and $n$ denotes the total number of genes.

The expression level was captured using TPM values, since we are quantifying the relative abundance of mRNA molecules per gene rather than the count of sequencing fragments. To test the stability of the TAI profiles and reduce the variance in the highly expressed genes[23,43], we performed several RNA-seq data transformations on the expression matrices: square-root transformation (used for the main figures), log transformation with a pseudo-count of 1 ($\log_2(\text{TPM} + 1)$), 'regularized log' transformation[91] (rlog), and rank transformation (that is, genes were ranked by level of expression at each stage). To reduce potential outliers, the median abundance value of replicates was chosen to represent the expression level ($e_{is}$).

The statistical significance of the resulting profiles was assessed using non-parametric permutation tests (flat line test, reductive hourglass test and one-sided pairwise TAI test), using the FlatLineTest(), ReductiveHourglassTest() and PairwiseTest() functions implemented in myTAI[41]. The $P$ value defines (for each tested shape) the probability that the observed TAI pattern is drawn from a random set of TAI profiles with permuted gene ages. We defined 'early' stages as S1–2, 'mid' as S3–4, and 'late' as S5 in *F. serratus*, and 'early' stages as S1–4, 'mid' as S4.5, and 'late' as S5 in *F. distichus*, due to differences in developmental stage correspondence. These tests, including those for sex differences, were performed with 50,000 permutations.

For the pTAI analysis, we used the function pMatrix() implemented in myTAI[41], which calculates the contribution of each gene to the TAI at each stage by multiplying the phylostratum of each gene by its expression level divided by the sum of expression of all genes, that is,

$$\text{pTAI}_{is} = \frac{ps_i \cdot e_{is}}{\sum_{i=1}^{n} e_{is}}$$

where, like TAI, $p_{si}$ denotes the relative gene age (phylostratum) for a given gene $i$ and $e_{is}$ denotes the expression level of a given gene $i$ at developmental stage $s$ and $n$ denotes the total number of genes. The elbow method was used to identify 500 genes with the highest TAI contribution in each developmental stage; genes driving the TAI value across all unicellular or multicellular stages were inferred via intersection. For consistency with the main TAI analyses, square-root transformation was applied before the pTAI analysis.

### Transcriptome specificity index

To investigate whether the transcriptome at the waist of the hourglass is composed of broadly expressed genes compared to other stages, we first indexed each gene by its relative expression specificity/breadth across development using tau[48,49], that is,

$$\text{tau}_i = \frac{\sum_{i=1}^{N} (1 - \hat{e}_i)}{N - 1}; \hat{e}_i = \frac{e_i}{\max_{1 < i < n}(e_i)}$$

where $N$ is the number of stages, $\hat{e}_i$ is the expression level of a given gene $i$ normalized by the maximal expression value. A lower tau indicates low stage-specificity (in other words, broad expression), and vice versa. The resulting tau values across all genes are stratified into deciles (tau-stratum), which enables analogous comparisons to TAI. It should be noted that PS and tau are not correlated (Kendall's $\tau \approx 0.05$ in both *Fucus* species), indicating that these metrics capture independent signals. In contrast to the TAI, the TSI captures the average expression specificity/breadth of a given transcriptome, weighted by the expression level of each gene, that is,

$$\text{TSI}_s = \sum_{i=1}^{n} \left( \frac{\text{ts}_i \cdot e_{is}}{\sum_{i=1}^{n} e_{is}} \right)$$

where $ts_i$ denotes the relative expression specificity/breadth (tau-stratum) for a given gene $i$, $e_{is}$ denotes the expression level of a given gene $i$ at developmental stage $s$ and $n$ denotes the number of genes. The median abundance of replicates was chosen to represent the expression level ($e_{is}$). Existing functions in myTAI were repurposed for this analysis.

### Transcriptome divergence index

To explore whether unicellular stages in *Ectocarpus* not only exhibited a young transcriptome, but also genes under relaxed purifying selection, we computed the TDI. In contrast to the TAI, the TDI captures the average gene selective pressure (divergence-stratum; based on deciled dN/dS ratios) of a given transcriptome, weighted by the expression level of each gene. The divergence-stratum of each gene in *Ectocarpus* was inferred from dN/dS ratios using orthologr[43]. In brief, one-to-one orthologues were inferred between *Ectocarpus* sp. 7 and *Ectocarpus subulatus*[92], using best reciprocal hits, and the dN/dS ratio was computed using the default "Comeron" estimation method. Importantly, >99% of one-to-one orthologue comparisons fell below the dN/dS ratio of 1, indicating that we are quantifying the degree of purifying selection. Next, the resulting dN/dS ratios across all genes are stratified into deciles, with the scale ranging from 1 (strong purifying selection) to 10 (weakest purifying selection). *Fucus* species were precluded from this analysis due to the short divergence time between *F. serratus* and *F. distichus*, approx. 4 million years ago[93], resulting in more than 10% of genes having dN/dS of 0. For genes with more than one isoform, the divergence-stratum of the oldest isoform was used.

Using myTAI[41], we calculated the TDI for each stage as follows,

$$\text{TDI}_s = \sum_{i=1}^{n} \left( \frac{\text{ds}_i \cdot e_{is}}{\sum_{i=1}^{n} e_{is}} \right)$$

where $ds_i$ denotes the relative divergence level (divergence-stratum) for a given gene $i$. The term $e_{is}$ denotes the expression level of a given gene $i$ at developmental stage $s$. The median abundance of replicates was chosen to represent the expression level ($e_{is}$).

### Distance/similarity-based transcriptome comparison

To quantify the overall distance/similarity between the transcriptomes of embryo stages in the two *Fucus* species, we computed the Pearson correlation, Spearman correlation, Manhattan distance and Jensen–Shannon distance (JSD) metric. Several metrics were employed owing to issues of calculating distance in high-dimensional data[94]. To compare expression levels between species, we compared the expression levels (abundance) of orthogroups (sets of orthologues and paralogues) using OrthoFinder v2.5.4[95], treating genes as isoforms and orthogroups as genes when importing the RNA-seq data using tximport v1.26.1[86]. Note, expression was quantified using length-normalized TPM to avoid bias from different gene lengths between species. Orthogroups were used rather than one-to-one orthologues (inferred through procedures such as best reciprocal hit), since orthogroups also capture the expression profile of in-paralogues, thereby covering more genes in the genome. The abundance data was transformed using rlog[91]. Correlation matrices were computed using cor() from the stats package in R[96], while Manhattan and JSD metrics were computed using the R package philentropy v0.7.0[97]. We employed the same approach (using log-transformed orthogroup abundance) to compare the overall transcriptome distance/similarity between life cycle stages in *Ectocarpus* and embryo stages in the two *Fucus* species, as well as to determine the corresponding stages between an early embryo stage of *L. digitata* sporophyte and each embryo stage of *Fucus*.

### Enrichment analyses

To explore gene function, GO terms were obtained using InterProScan v5.61-93.0[98]. GO enrichment analysis was then performed on genes contributing most to *Ectocarpus* TAI (inferred from the partial TAI value

of each individual gene, pTAI$_i$) using Fisher's exact test statistics with the parent–child algorithm as implemented in TopGO v2.48.0[99,100]. Statistical tests and significance levels are indicated in the text and figure legends.

## Reporting summary

Further information on research design is available in the Nature Portfolio Reporting Summary linked to this article.

## Data availability

Data are available in NCBI Bioproject under accession PRJNA1090323. Further sample details and accession codes are available in Supplementary Table 1.

## Code availability

The analysis code is available at https://github.com/LotharukpongJS/hourglass_brownalga.

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

**Acknowledgements** The authors thank C. Martinho, A. Lipinska, J. Tan Shen Yi and J. Barrera-Redondo for insightful discussions; F. Weinberger for providing *F. distichus* samples; N. Kalábová for implementing multithreaded permutation tests in myTAI; and D. Weigel for support and sponsorship of H.-G.-D. This work was supported by the MPG, the ERC (grant no. 864038 to SMC) and the BMBF-funded de.NBI Cloud within the German Network for Bioinformatics Infrastructure (de.NBI) (031A532B, 031A533A, 031A533B, 031A534A, 031A535A, 031A537A, 031A537B, 031A537C, 031A537D and 031A538A). S.M.C. is supported by the Moore Foundation (GBMF11489) and the Bettencourt-Schuller Foundation. H.-G.D. is supported by a Royal Society Wolfson Fellowship (RSWF\R1\241004). Computations were also performed in the MPCDF Cobra supercomputer in Garching, Germany and the cluster of the Max Planck Campus in Tübingen, Germany. J.S.L. thanks the International Max Planck Research School 'From Molecules to Organisms'.

**Author contributions** J.S.L.: investigation (equal); formal analysis (lead); visualization (lead); writing, original draft (lead); writing, review and editing (supporting). M.Z.: investigation (equal); methodology (supporting). R.L.: investigation (equal); methodology (supporting); visualization (supporting). D.L.: investigation (supporting); methodology (supporting); visualization (supporting). H.-G.D.: conceptualization (equal); funding acquisition (supporting); data curation (equal); methodology (lead); visualization (supporting); project administration (equal); supervision (equal); writing, original draft (equal); writing, review and editing (supporting). S.M.C.: conceptualization (lead); funding acquisition (lead); methodology (equal); project administration (lead); supervision (equal); visualization (supporting); writing, original draft (supporting); writing, review and editing (lead).

**Funding** Open access funding provided by Max Planck Society.

**Competing interests** The authors declare no competing interests.

**Additional information**
**Correspondence and requests for materials** should be addressed to Hajk-Georg Drost or Susana M. Coelho.

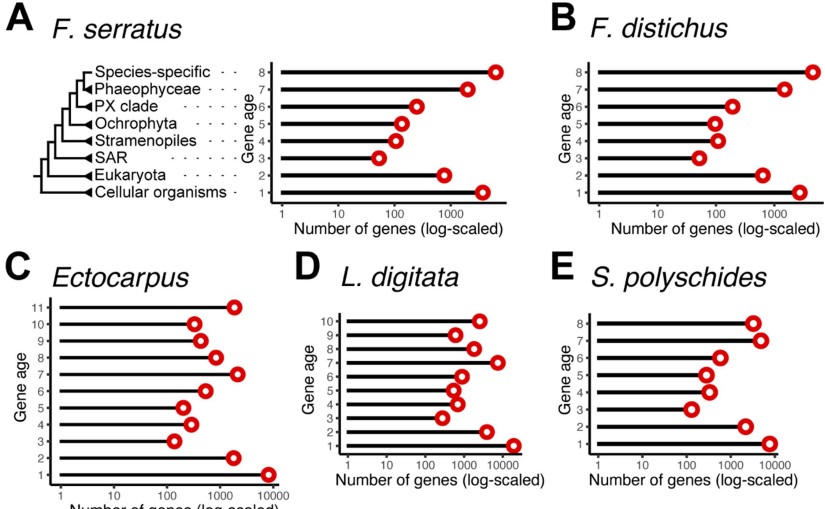

**Extended Data Fig. 1 | Summary statistics for the phylostratigraphic maps of *F. serratus*, *F. distichus*, *Ectocarpus* (species 7), *L. digitata* and *S. polyschides*.** Taxonomic ranks and the number of genes assigned to each rank (phylostratum [PS] - gene age ranks) in brown algae: **A**, *F. serratus*; **B**, *F. distichus*; **C**, *Ectocarpus*; **D**, *L. digitata*; and **E**, *S. polyschides*. Evolutionarily younger genes have a higher gene age rank and *vice versa* for evolutionarily older genes. Lineage information is detailed in *F. serratus* (**A**), and is the same across all five species for PS1 (cellular organisms) to 7 (Phaeophyceae; the evolutionary transition to complex multicellularity).

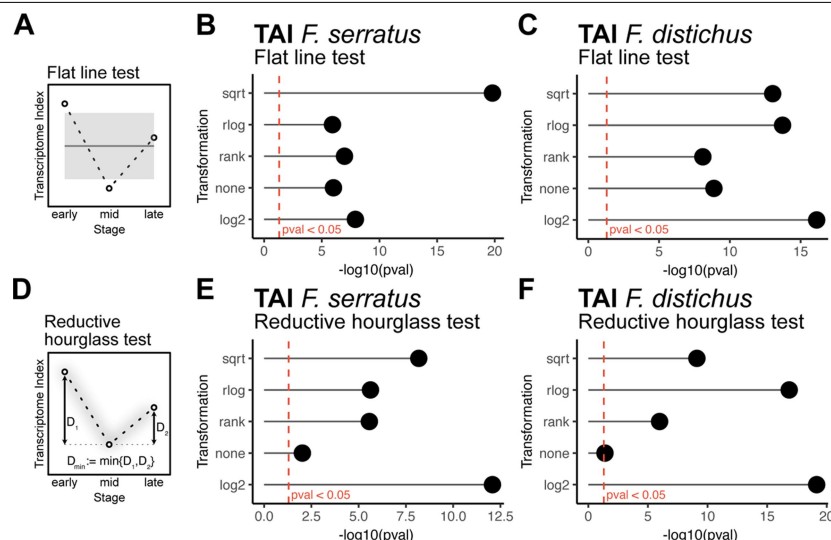

**Extended Data Fig. 2 | Permutation tests on the transcriptome evolution patterns in *Fucus* embryogenesis under various RNA-seq transformations.** Previous studies have shown that RNA-seq transformations can affect TAI profiles[23,47], motivating this analysis on previously reported and additional transformations: identity ('none'), square-root ('sqrt'), logarithmic ('log2'), as well as non-parametric ranking ('rank') and variance-stabilising ('rlog'). **A**, The flat line test evaluates any significant deviation of an observed transcriptome evolution pattern from a flat line[43]. **B,C**, Flat line tests in: **B**, *F. serratus* and **C**, *F. distichus*, under various RNA-seq transformations. **D**, The reductive hourglass test evaluates whether an observed transcriptome evolution pattern follows an hourglass (high-low-high) pattern, based on the statistical significance of the observed minimum difference ($D_{min}$) between the 'mid' module and the 'early' ($D_1$) or 'late' ($D_2$) modules. **E,F**, Reductive hourglass tests in **E**, *F. serratus* and **F**, *F. distichus*, under various RNA-seq transformations. All tests were conducted with 50,000 permutations and all resulting profiles were statistically significant (p-value < 0.05).

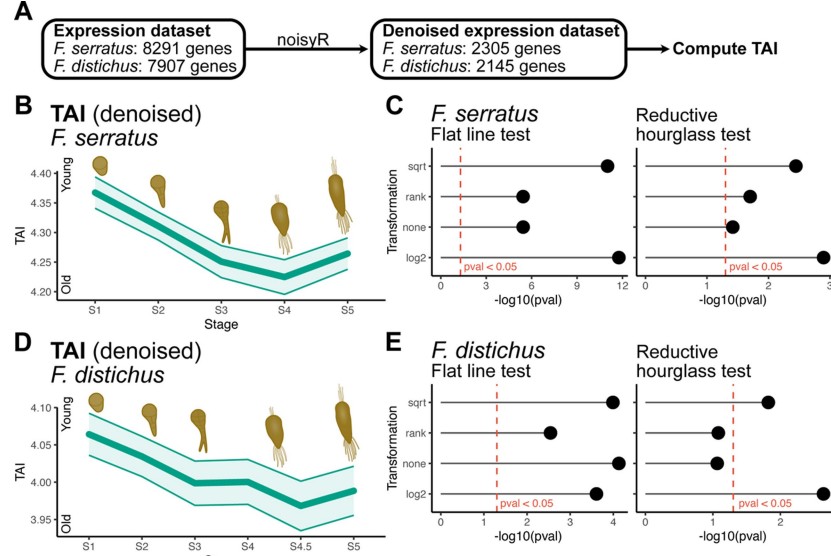

**Extended Data Fig. 3 | Transcriptome age patterns in *Fucus* development using a denoised dataset. A**, Overview of removal method for genes with 'noisy' expression profile using noisyR. **B**, TAI across embryogenesis in *F. serratus* using a denoised dataset. **C**, Flat line and reductive hourglass permutation tests in *F. serratus* under various RNA-seq transformations. **D**, TAI across embryogenesis in *F. distichus* using a denoised dataset. **E**, Flat line and reductive hourglass permutation tests in *F. distichus* under various RNA-seq transformations. 50,000 bootstraps were used to compute the standard deviation in **B** and **D**. Permutation tests were performed with 50,000 permutations.

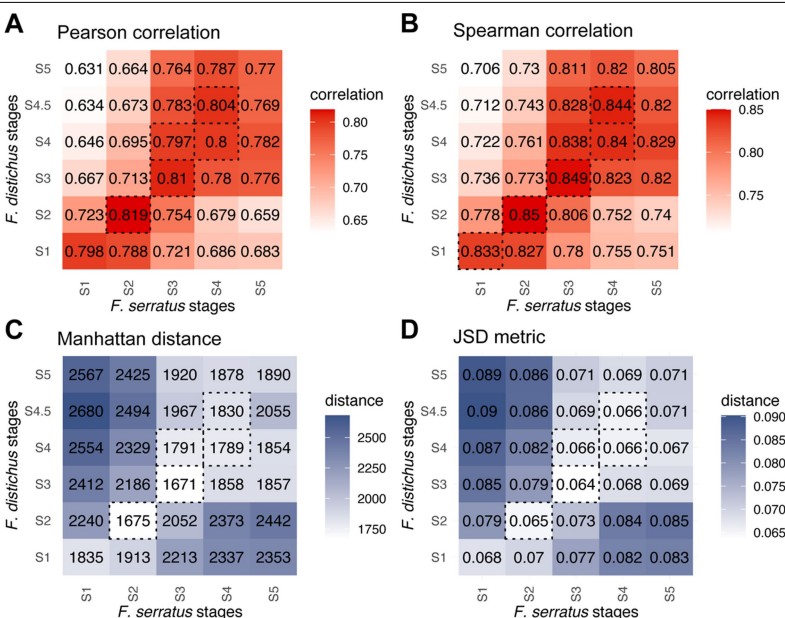

**Extended Data Fig. 4 | Transcriptome comparisons between embryo stages of *Fucus* species and corresponding embryo stages.** Expression levels of shared orthogroups were compared rather than one-to-one orthologues to mitigate the effect of in-paralogues. Shown are the median: **A**, Pearson correlation; **B**, Spearman correlation; **C**, Manhattan distance; and **D**, Jensen-Shannon Distance (JSD) metric between each embryonic stage under the rlog transformation. Dashed boxes indicate the five most similar/closest comparisons between the two closely related species.

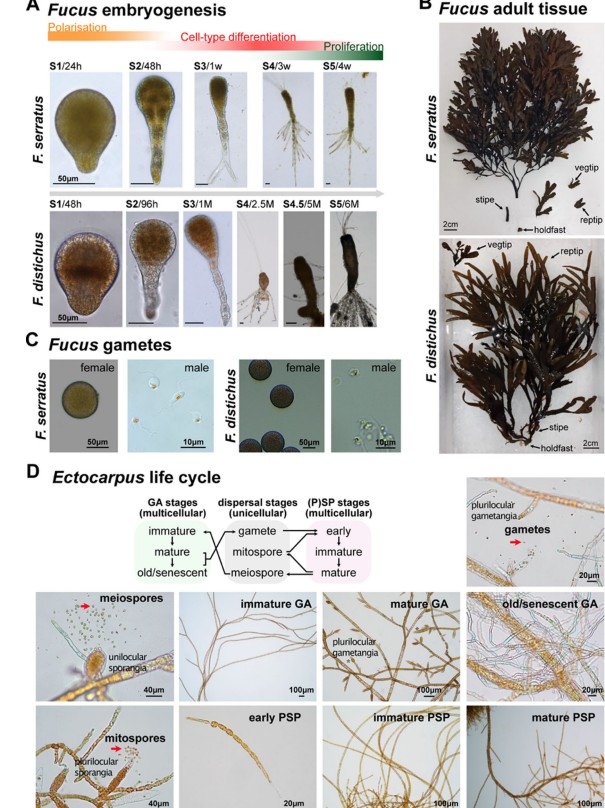

**Extended Data Fig. 5 | Images of life cycle stages of *Fucus* and *Ectocarpus*.**
**A**, Embryo stages of *F. serratus* and *F. distichus*. In both species, the embryos go through phases of polarization, cell-type differentiation and proliferation of established cell types. **B**, Adult tissues of *F. serratus* and *F. distichus*. **C**, Gametes of *F. serratus* and *F. distichus*. **D**, *Ectocarpus* life cycle stages with a scheme to link the sampled stages. 'GA' and '(P)SP' denote multicellular gametophytes and (partheno)sporophytes, respectively. The red arrows point to unicells.

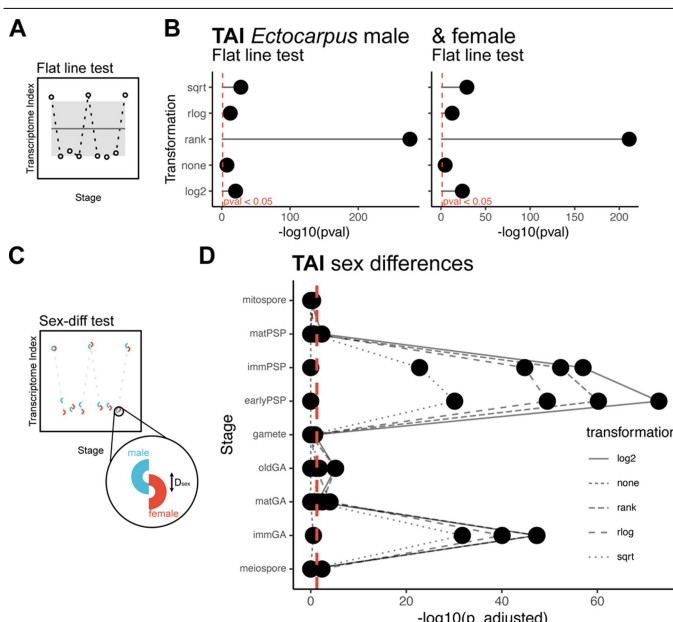

**Extended Data Fig. 6 | Permutation tests on the transcriptome evolution patterns in *Ectocarpus*. A**, The flat line test evaluates any significant deviation of an observed transcriptome evolution pattern from a flat line. **B**, Flat line tests in male and female *Ectocarpus*, under various RNA-seq transformations. All profiles were statistically significant. **C**, The two-sided sex difference permutation test evaluates sex differences ($D_{sex}$) in the transcriptome evolution pattern at a given stage. **D**, Differences in TAI between sexes at each developmental stage in *Ectocarpus*, under various RNA-seq transformations. The resulting p-values were adjusted using Bonferroni correction for multiple testing. All tests were conducted with 50,000 permutations.

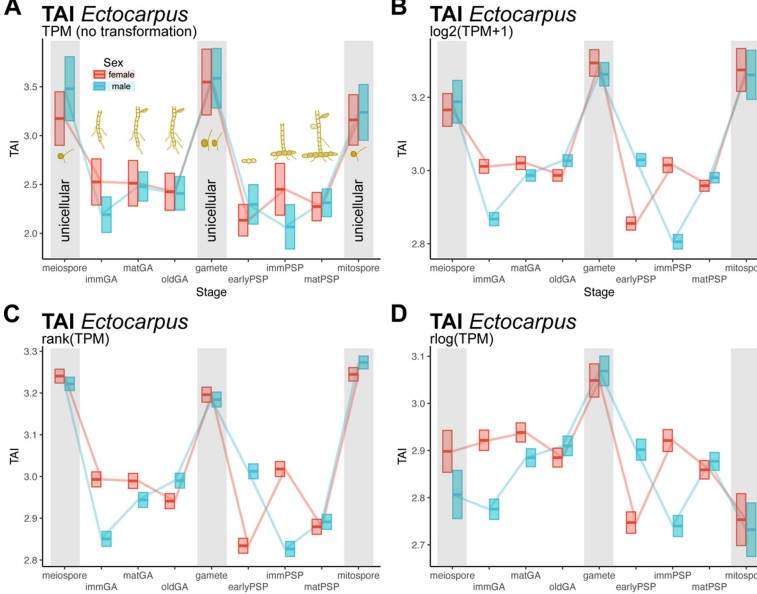

**Extended Data Fig. 7 | TAI profile across life cycle stages of *Ectocarpus*, under different RNA-seq transformations.** TAI was computed after applying different RNA-seq transformations: **A**, raw TPM (none); **B**, log2(TPM + 1); **C**, rank; and **D**, regularized log. A lower TAI marks a transcriptome composed of evolutionarily older genes and *vice versa*. Blue lines represent male stages while red lines represent female stages. The upper and lower limits of the boxes demarcate the s.d., based on 50,000 bootstraps.

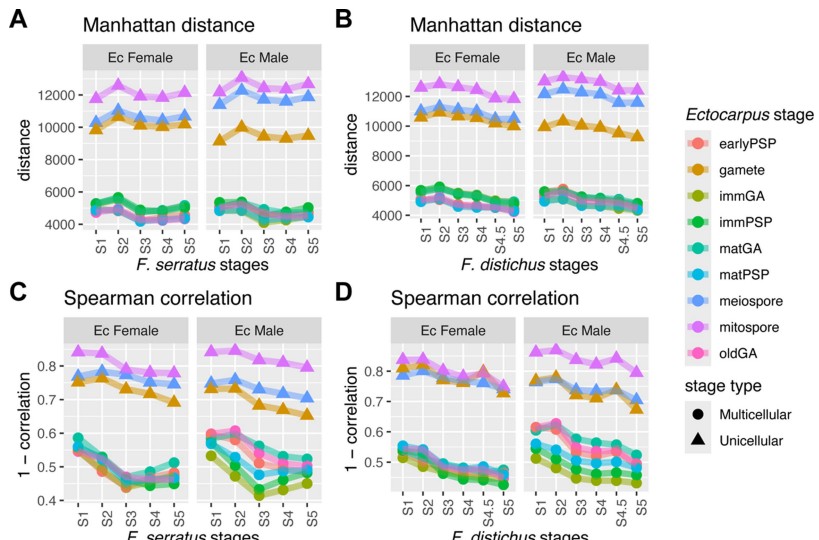

**Extended Data Fig. 8 | Transcriptome comparisons between *Ectocarpus* and *Fucus* reveal expression-level conservation of multicellular stages.** Expression levels of shared orthogroups were compared rather than one-to-one orthologues to mitigate the effect of in-paralogues. **A**,**B**, Manhattan distance between **A**, *F. serratus* or **B**, *F. distichus* against the life cycle stages in female and male *Ectocarpus* using log-transformed orthogroup abundance. **C**,**D**, Similar analysis using Spearman correlation for **C**, *F. serratus* or **D**, *F. distichus*, to present monotonic relationship. Line colour marks distinct *Ectocarpus* stages, whose type (i.e., multicellular or unicellular) is denoted by the point shape.

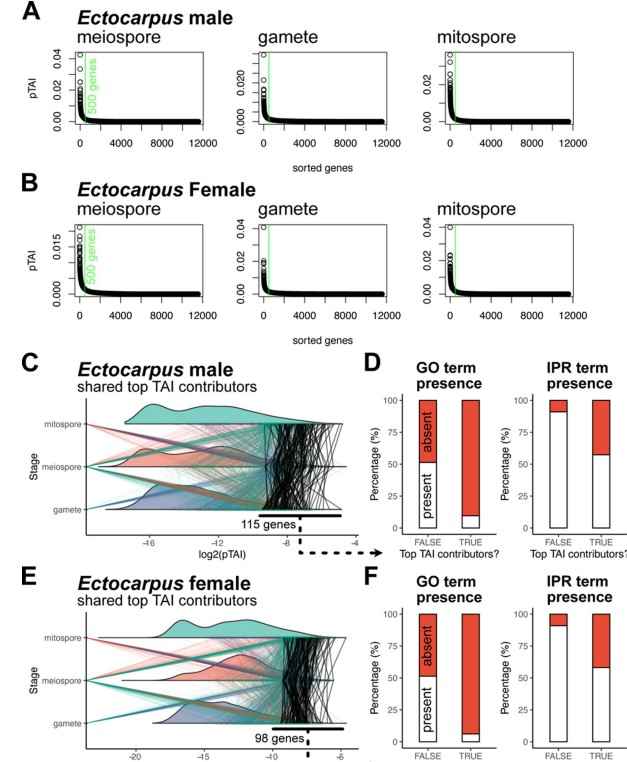

**Extended Data Fig. 9 | pTAI in unicellular stages of *Ectocarpus*. A,B,** Genes
ranked by their pTAI across each unicellular stage of **A**, male and **B**, female
*Ectocarpus*. The green line indicates the top 500 gene cutoff based on the elbow
method, i.e., genes that explain most of the overall TAI ('top contributors').
**C**, Shared top contributors in unicellular stages of male *Ectocarpus*. The coloured
lines denote the pTAI value of a given top-contributor gene in one unicellular
stage mapped across other unicellular stages. Shared top contributors across
all stages are denoted by black lines. **D**, The presence/absence of gene ontology
(GO) and interproscan (IPR) domain terms in the shared top contributors
in unicellular stages of *Ectocarpus*. **E,F**, The same analysis done on female
*Ectocarpus*. For visual clarity, the pTAI values were log-transformed.

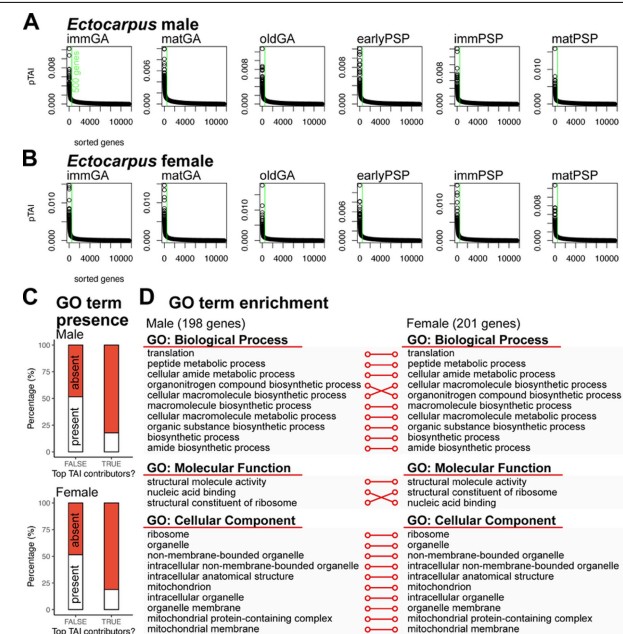

**Extended Data Fig. 10 | pTAI in multicellular stages of *Ectocarpus*.**
**A**,**B**, Genes ranked by their pTAI across each multicellular stage of **A**, male and **B**, female *Ectocarpus*. The green line indicates the top 500 gene cutoff based on the elbow method, i.e., genes that explain most of the overall TAI ('top contributors'). **C**, The presence/absence of gene ontology (GO) terms in the shared top contributors in multicellular stages of *Ectocarpus* (198 in males and 201 in females). **D**, The relationship between ten most significant GO terms ranked by p-values for each GO category between male and female *Ectocarpus*. Significance is defined by stringent Bonferroni correction which justifies padj <0.1 as significance threshold.

**A** *L. digitata* life cycle

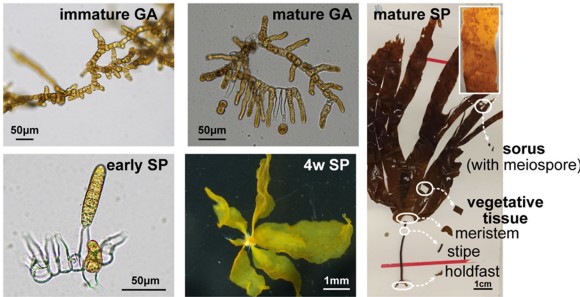

**B** *S. polyschides* life cycle

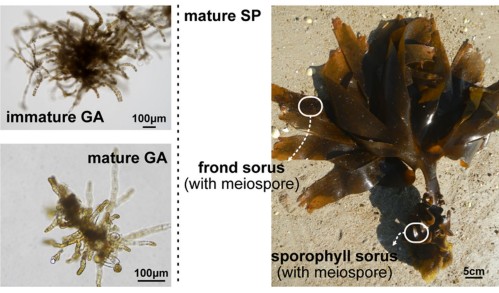

**Extended Data Fig. 11 | Images of life cycle stages of kelps *Laminaria digitata* and *Saccorhiza polyschides*. A**, Sampled life cycle stages of *L. digitata*. **B**, Sampled life cycle stages of *S. polyschides*. Note, in *S. polyschides*, publicly available data from ref. 71 was used for immature GA. 'GA' and 'SP' denote multicellular gametophytes and sporophytes, respectively.

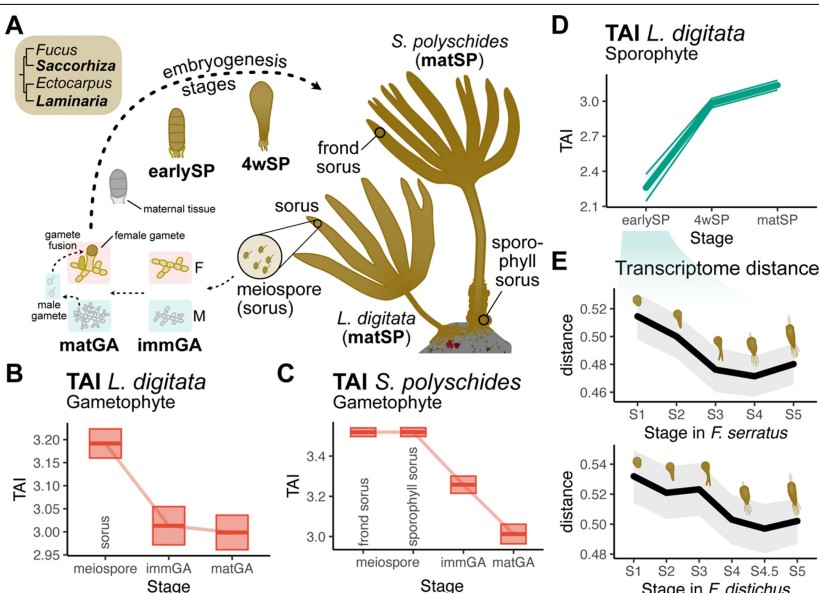

**Extended Data Fig. 12 | TAI profile across life cycle stages of *Laminaria digitata* and *Saccorhiza polyschides*. A**, Outline of phylogenetic position and life cycle in *L. digitata* and *S. polyschides*. Note that earlier embryogenesis stages could not be sampled due to potential contamination from maternal gametophytic tissue. Stages not included are in grey-scale. **B**, TAI profile across gametophyte development in *L. digitata*. **C**, TAI profile across gametophyte development in *S. polyschides*. In both **B** and **C**, the unicellular meiospore-enriched sori were used as a proxy for meiospores. **D**, TAI profile across sporophyte development in *L. digitata*. The mature SP stage is represented by vegetative tissues. **E**, Transcriptome distance between an early sporophyte stage in *L. digitata* and the embryo stages of *Fucus*, using the Manhattan distance of normalized log-transformed orthogroup abundance. 'GA' denotes gametophyte (n) and 'SP' denotes sporophytes (2n). 'imm' and 'mat' denote sexually immature and mature stages, respectively. 50,000 bootstraps were used to compute the s.d.

# Reporting Summary

## Statistics

For all statistical analyses, confirm that the following items are present in the figure legend, table legend, main text, or Methods section.

| n/a | Confirmed | |
|---|---|---|
| ☐ | ☒ | The exact sample size ($n$) for each experimental group/condition, given as a discrete number and unit of measurement |
| ☒ | ☐ | A statement on whether measurements were taken from distinct samples or whether the same sample was measured repeatedly |
| ☐ | ☒ | The statistical test(s) used AND whether they are one- or two-sided *Only common tests should be described solely by name; describe more complex techniques in the Methods section.* |
| ☐ | ☒ | A description of all covariates tested |
| ☐ | ☒ | A description of any assumptions or corrections, such as tests of normality and adjustment for multiple comparisons |
| ☐ | ☒ | A full description of the statistical parameters including central tendency (e.g. means) or other basic estimates (e.g. regression coefficient) AND variation (e.g. standard deviation) or associated estimates of uncertainty (e.g. confidence intervals) |
| ☐ | ☒ | For null hypothesis testing, the test statistic (e.g. $F$, $t$, $r$) with confidence intervals, effect sizes, degrees of freedom and $P$ value noted *Give P values as exact values whenever suitable.* |
| ☒ | ☐ | For Bayesian analysis, information on the choice of priors and Markov chain Monte Carlo settings |
| ☒ | ☐ | For hierarchical and complex designs, identification of the appropriate level for tests and full reporting of outcomes |
| ☐ | ☒ | Estimates of effect sizes (e.g. Cohen's $d$, Pearson's $r$), indicating how they were calculated |

*Our web collection on statistics for biologists contains articles on many of the points above.*

## Software and code

Policy information about availability of computer code

| Data collection | Codes are available at https://github.com/LotharukpongJS/hourglass_brownalga |
|---|---|
| Data analysis | Data analysis codes are available at  https://github.com/LotharukpongJS/hourglass_brownalga |

For manuscripts utilizing custom algorithms or software that are central to the research but not yet described in published literature, software must be made available to editors and reviewers. We strongly encourage code deposition in a community repository (e.g. GitHub). See the Nature Portfolio guidelines for submitting code & software for further information.

## Data

Policy information about availability of data

All manuscripts must include a data availability statement. This statement should provide the following information, where applicable:
- Accession codes, unique identifiers, or web links for publicly available datasets
- A description of any restrictions on data availability
- For clinical datasets or third party data, please ensure that the statement adheres to our policy

All data is in a repository, accession codes are provided in the manuscript (PRJNA1090323). Biological material is available in Culture Collections (RCC) or upon request.

## Research involving human participants, their data, or biological material

Policy information about studies with human participants or human data. See also policy information about sex, gender (identity/presentation), and sexual orientation and race, ethnicity and racism.

| | |
|---|---|
| Reporting on sex and gender | n.a |
| Reporting on race, ethnicity, or other socially relevant groupings | n.a |
| Population characteristics | n.a |
| Recruitment | n.a |
| Ethics oversight | n.a |

Note that full information on the approval of the study protocol must also be provided in the manuscript.

## Field-specific reporting

Please select the one below that is the best fit for your research. If you are not sure, read the appropriate sections before making your selection.

☐ Life sciences    ☐ Behavioural & social sciences    ☒ Ecological, evolutionary & environmental sciences

For a reference copy of the document with all sections, see [nature.com/documents/nr-reporting-summary-flat.pdf](https://nature.com/documents/nr-reporting-summary-flat.pdf)

## Ecological, evolutionary & environmental sciences study design

All studies must disclose on these points even when the disclosure is negative.

| | |
|---|---|
| Study description | We studied the evolutionary transcriptome of several brown algal species during development |
| Research sample | Samples are five brown algal species (Fucus distichus, Fucus serratus, Ectocarpus species 7, Laminaria digitata, Saccorhiza polyschides). |
| Sampling strategy | n.a |
| Data collection | RNA was extracted from three biological replicates, across several developmental stages in the life cycle |
| Timing and spatial scale | Discrete stages of the life cycle were analysed |
| Data exclusions | no data was exclluded |
| Reproducibility | replicate samples were used. More details can be found in the source code |
| Randomization | n.a |
| Blinding | n.a |

Did the study involve field work?    ☐ Yes    ☒ No

## Reporting for specific materials, systems and methods

We require information from authors about some types of materials, experimental systems and methods used in many studies. Here, indicate whether each material, system or method listed is relevant to your study. If you are not sure if a list item applies to your research, read the appropriate section before selecting a response.

## Materials & experimental systems

| n/a | Involved in the study |
|---|---|
| ☒ ☐ | Antibodies |
| ☒ ☐ | Eukaryotic cell lines |
| ☒ ☐ | Palaeontology and archaeology |
| ☒ ☐ | Animals and other organisms |
| ☒ ☐ | Clinical data |
| ☒ ☐ | Dual use research of concern |
| ☒ ☐ | Plants |

## Methods

| n/a | Involved in the study |
|---|---|
| ☒ ☐ | ChIP-seq |
| ☒ ☐ | Flow cytometry |
| ☒ ☐ | MRI-based neuroimaging |

## Plants

| Seed stocks | na |
|---|---|
| Novel plant genotypes | na |
| Authentication | na |

