## [Peer Review file · Nature]

Manuscript Title: A transcriptomic hourglass in brown algae

Reviewer Comments & Author Rebuttals

Reviewer Reports on the Initial Version:

Referees' comments:

Referee #1 (Remarks to the Author):

Lotharukpong et al set out to explore transcriptome conservation patterns in brown algae and specifically test if the developmental hourglass previously described in plants, animals and controversially in fungi is tied to embryogenesis or to complex development in general? This is a great idea and if confirmed could contribute to solving a long-standing debate and explain why the hourglass model has been accepted more in certain lineages (e.g. animals) whereas debated in others, where embryogenesis is missing (fungi). This is a well-written and resource-rich study conducted with high methodological rigor (with some exceptions, see below). The complex analyses are well-presented in an easy-to-follow narrative and the independent evolution of embryogenesis in brown algae offers a good model system.

However, I think the ms currently falls short in testing whether the hourglass is tied to embryogenesis or complex development, due to sampling and analytical issues explained below. If the Authors really want to test this then they should test the presence of an hourglass in embryogenetic (done) and multiple multicellular but not embryogenetic processes. A systematic analysis for the latter is missing, only 1 dataset was analyzed. I think this constrains the generality of the findings. This could be remedied by analyzing more nonembryogenic multicellular developmental processes.

Due to the above mentioned caveat, it is not clear how the Authors arrived to the conclusion Abstract/"Our findings suggest that transcriptome conservation in brown algae is associated with cell differentiation stages, but not necessarily linked to embryogenesis.". If anything, it looks like it is linked to embryogenesis, though clarity on whether a phase of cell differentiation exists in the comparator species (*Ectocarpus*) and the apparent lack of an hourglass in this taxon makes this conclusion confusing.

Results - it is great and important information that *Fucales* species can be sampled without contamination from parental tissues. What is the situation in *Ectocarpus*? Given the importance of this species in the comparisons it may be important to note how this species can be sampled.

Results/"using tau as a proxy for pleiotropy" - I challenge using tau as a proxy. Tau measures to what extent a gene is 'housekeeping' or 'tissue/stage-specific'. Pleiotropy refers to alternative (not constantly needed/rarely needed) functions of a given gene. I doubt tau can substitute more sophisticated measures of pleiotropy available .e.g in yeast based on experimental data. Therefore, I agree with the conclusion "were represented by more broadly expressed genes (i.e., low tau)

whereas early and later developmental stages were characterised by more stage-specific genes”, but not with the next sentence (Therefore, more pleiotropic genes are associated...). The references provided do not support the usage of tau as a substitute of pleiotropy.

Results/Fucus TAI profiles - the methods mentions the flatline and reductive hourglass tests for assessing the significance of the hourglass shape. I think should be mentioned here, in the main text what p-values did these tests give for Fucus.

Results/Fucus TAI profiles - An uncertainty here is that in *F. distichus* the ‘waist’ of the hourglass is stage 4.5, which is not defined anywhere in the text (in particular, missing from Figure S5 defining the stages). What does this stage correspond to, was it obtained by some extrapolation from stages 4 and 5?

Throughout the ms the term “evolutionary transcriptomes” seems superfluous. The Authors talk about transcriptomes, I don’t see how a transcriptome could be ‘evolutionary’. Please consider rephrasing.

Throughout the text, please provide exact p-values, not cutoffs (e.g. <0.05).

“In *F. serratus* males, different tissues presented significantly different TAI values (flat line test; $p < 0.001$)” - The flat line test might not be an appropriate test here. The Authors are basically comparing age distributions of expression weighted gene ages. I suggest comparing transcriptomes in a pairwise manner. The flatline test can be misleading if only a single transcriptome differs from the rest. An example to this: “This difference was also observed in *F. distichus* (Fig. 3C)” - Based on Fig3C only *vegtip* seems to differ, all others look the same. Also, could vegetative and reproductive tissues be clearly marked?

“Multicellularity constrains transcriptome evolution in *Ectocarpus*” - My concern with this chapter is that it does not explicitly look for an hourglass, as it should, based on the aim stated. The chapter shows that multicellular stage transcriptomes are more conserved than those of unicellular stages. How does this result relate to an hourglass? Is there an hourglass during gametophyte and/or partheno-sporophyte development? Why are unicellular stages involved in the analysis? Sperms and eggs were not involved in tests of animal/plant hourglasses either, so it feels the analyses should be restricted to stages of the multicellular structure. Or should these analyses be not viewed as an attempt to find an hourglass? Maybe there was an hourglass if only multicellular stages are considered?

In *Fucus*, “conservation was coupled to the transition from cell differentiation to proliferation”. Are there homologous/analogous steps during multicellular development of *Ectocarpus*? What happens before the establishment of the immature gametophyte stage? Is there a stage, like early colony formation, focusing on differentiation, followed by largely just growth? Could those stages be sampled? I think the essence of the comparison lies in identifying and testing the developmental period of *Ectocarpus* that may show signs of differentiation and proliferation (maybe the early stages of the establishment of the colony?). The current dataset in my opinion is not going to depths for testing this and uses a suboptimal sample set. It would be important for such a key conclusion to

not hinge on incomplete exploration of the sample space.

Fig4 please defined abbreviations

Though I'm not a brown alga expert, it looks like *Fucus* and *Ectocarpus* all have gametes and multicellular gametophyte stages. That is, a comparison like Fig 4 could be performed in *Fucus* also and it could be tested how the pattern there relate to the hourglass in embryogenesis. This would contribute to the more thorough testing I miss for substantiating the main conclusion.

Discussion/"pointing to the redeployment of pre-existing molecular toolkit for brown algal complex multicellularity" - this is a fashionable and abstract idea these days, but in this specific context, with a stramenopile ancestry, what pre-existing toolkit should be envisioned to that could underpin the body plan? Isn't it possible that simply changes in the expression regulation of conserved genes represent the innovation underlying multicellularity? This scenario is less abstract, but offers an easier to grasp explanation, that is, simply innovation but not in gene content (as is predicted by the cis-regulatory hypothesis).

Discussion/"from a cell-type differentiation stage where the body plan of the adult *Fucus* is established, to a proliferative stage" - this goes back to my arguments on sampling. If this is the case, then an effort should be made to identify differentiation to proliferation switches in *Ectocarpus* and the hourglass should be tested there.

Discussion/"We reason that constraints on morphology are imposed continuously by regulatory complexity that results from dynamic and tissue-specific transcriptional controls, resulting in a conserved transcriptome." - please check sentence, too abstract and/or makes little sense

Discussion/"this pattern may represent a fundamental characteristic of multicellular complex development across the entire eukaryotic tree of life" - This conclusion seems to be in conflict with the finding that there was an hourglass in *Fucus* embryogenesis but not in *Ectocarpus*. If this conclusion holds then why does *Ectocarpus* not have an hourglass? I accept that the presence of the hourglass in other taxa partially supports this, but then its not a conclusion drawn from these data. It should be noted though that in previous studies a clear distinction between complex development and embryogenesis was not made - the key good idea in this paper.

Methods/Distance/similarity-based transcriptome comparison - here, the Authors make expression comparisons across species. They primarily use TPM for quantifying expression, which is not normalized for unit gene/transcript length. As a result, if 1-1 orthologous genes have length differences between the compared species, this analysis will provide misleading results. The Authors should use length-normalized expression metrics (e.g. FPKM) or demonstrate that gene length differences are not affecting their results.

Methods/enrichment - how was 'contributing most' defined?

In supplementary excel files columns and abbreviations are not explained. Legends should be included.

Referee #2 (Remarks to the Author):

Review of "A Transcriptomic Hourglass in Brown Algae" by Lotharukpong et al.

In this paper, the authors show that the developmental hourglass model, previously established in animals, plants and fungi, also extends to brown algae- another clade of complex multicellularity. This is a novel and significant paper with important implications for our understanding of the evolution of multicellular development. From what I can tell, the data and analyses are sound, and the paper is well written. My main concern is that it is effectively two phylogenetically independent data points (one simple, and one complex multicellular brown algae lineage), which may not be robust.

Summary of key results: First, the authors demonstrate that two species of complex multicellular brown algae, *Fucus serratus* and *Fucus distichus*, exhibit a transcriptomic hourglass pattern during development. The 'waist' of the hourglass, representing the stage with the greatest conservation in the transcriptome, is mainly expressing old genes (things that pre-date the evolution of brown algae themselves), and the repression of young genes. The waist of the hourglass corresponds to the portion of time in which the embryo is defining its body plan, which makes sense, as this is arguably the most critical stage in development that is under the greatest evolutionary constraint.

Second, the authors show that in the filamentous brown alga *Ectocarpus*, which lacks a embryogenesis and instead grows in a modular manner, the transcriptome is most conserved in multicellular stages, compared to a unicellular stage. This shows that transcriptional constraints are not simply a feature of embryogenesis, but the life history stage in which morphological patterning is being determined.

Finally, the authors show that male reproductive tissues are enriched in young genes in *Fucus*, mirroring the "out of testis" pattern seen in animals and plants. I'm less sure what to make of these results, as I am less familiar with the literature on the evolution of sex-specific tissues.

The developmental hourglass model has been intensely studied in animals, with some recent work in plants and fungi. This is the first test of the model in the brown algae, which independently evolved complex multicellularity. The finding that brown algae also exhibit a developmental hourglass is significant, as it suggests this may be a universal feature of complex multicellularity and the constraints that come from the developmental phase in which 3D body plans are defined. By comparing species with and without embryogenesis, the authors further provide new insight into the relative contributions of multicellularity and embryonic development to the hourglass pattern. I think that this work will be of broad interest to the evo-devo / major transition communities.

In terms of methodology, the authors generated high quality stage-specific RNA-seq data for three diverse brown algal species. They employed standard comparative genomic methods to infer the evolutionary age of each gene and computed standard metrics (TAI, TDI, TSI) to quantify transcriptome conservation across development. The statistical analyses, including permutation

tests and bootstrapping, seem appropriate to me, but this is getting a bit afield of my expertise.

I don't see any major flaws in this paper, though it does feel a bit small (i.e., data light) for a contemporary Nature paper. I think the largest weakness is that just two species represent complex multicellularity in the brown algae, and they are both members of the same genus. In a sense, they're not independent, as they recently shared a common ancestor (not sure about exact divergence time, but they're clearly closely related as the authors had to exclude them as being too similar: "Fucus species were precluded from this analysis due to the short divergence time between *F. serratus* and *F. distichus*, which meant that more than 10% of genes having dNdS of 0". So for all intents and purposes, there is one phylogenetically independent datapoint with *Fucus*, and one with *Ecocarpus*. I think that seeing a similar pattern in divergent species of complex brown algae (maybe a *Macrocystis*?) would lend a great deal of support to the broader claims. However, I'm not sure if this is feasible, as I do not work with these organisms. This is not a need to have, but I'd love to hear the authors thoughts on the feasibility of broadening the diversity of their sample.

Another suggestion (that is not a requirement!) is that recent single-cell RNA-seq studies in animals have shown that the waist of the hourglass corresponds to the emergence of phylum-specific cell type transcriptional programs (i.e., see [10.1126/science.aar5780](https://doi.org/10.1126/science.aar5780) and [10.1038/s41586-019-0933-9](https://doi.org/10.1038/s41586-019-0933-9)). Do we see anything similar here? I'd love to see an elaboration of the intersection of the gene families coordinating multicellular development and the temporal series of development.

Finally, this is not a suggestion for this paper, but for future work. There's growing evidence that evolution of non-coding cis-regulatory elements plays a key role in developmental evolution, often by modulating the expression of developmentally-important genes. Specifically, cis-regulatory changes in non-coding elements have been shown to modulate developmental genes and contribute to morphological evolution across various animal taxa. If, for example, developmental enhancers are slow evolving (as has been shown in *Drosophila*), then this might contribute to the hourglass pattern in brown algae. You could figure this out by CHIP-seq or ATAC-seq.

I found the paper relatively easy to read, once I had memorized what TAI, TDI, TSI were. I appreciated that the authors explained their conceptual meaning in the figure legends, and would appreciate even a bit more of this if possible. I thought the overall conceptually-driven approach to the writing was quite effective and clear.

Referee #2 (Remarks on code availability):

I did not review the code. I've never run this kind of bioinformatics analysis, so I'm not the right person to review it.

Referee #3 (Remarks to the Author):

Lotharukpong et al. present an excellent dataset on the topic of transcriptomic hourglass patterns in brown algae. Their overall findings are very similar to what was shown before for animals and plants,

at least for two of their study species (*Fucus*). The most exciting part of these findings is that the multicellular life cycles of brown algae evolved independently of those of plants and animals. This implies that there is a generality in these patterns, which establish them as one of the few general biological rules.

Given this very general finding, it would be great if the authors would discuss more the actual genes that are causing this average pattern. There is some discussion of "pleiotropic genes", but this definition is too general, since it seems to include both house-keeping genes, as well as ancient regulatory genes or genes required for cell-cell communication. It would be great if the authors could add some additional analyses on the subset of genes that are thought to have arisen in the context of animal/plant multicellularity. Are they the same that are also enriched in the "waist" of the *Fucus* hourglass stages?

I list a number of further more detailed comments below that could improve the paper. Given that there are no line numbers, I reproduce their text sections and add my comment after ">"

Introduction

With the rise of complex multicellular lineages, a major question is how developmental processes can accommodate evolutionary change. In other words, how can the benefits of evolutionary change be co-opted without disrupting intricate developmental processes?

> not sure what this statement is alluding to; could be omitted

Two models have been proposed to explain how conserved developmental processes...

> There is actually a third model, originally proposed by Darwin: the phylotypic stage is most shielded from the influence of the environment and there are therefore fewer triggers for evolutionary change - see explicit discussion of this point in: Artieri, C. G., Haerty, W. & Singh, R. S. Ontogeny and phylogeny: molecular signatures of selection, constraint, and temporal pleiotropy in the development of *Drosophila*. *BMC Biol.* 7, 42 (2009).

Figure 1b

> please explain the term "matSP"

Figure 1c

> it is not clear which phases are haploid and which are diploid

page5

...robust to all RNA-seq data transformations (Fig. S2)..

> given that there was substantial discussion on RNA-seq data transformation for studying hourglass patterns, it would be good to expand this section with more details. At the minimum, one should find a more extensive description in the legend of Fig. S2 (and not only hidden in the Methods section)

page5

Despite a shift in the timing of the developmental stages between the two species, the transcriptome of the most conserved stage of *F. distichus* was also the one most similar to the *F.*

serratus stage with highest degree of conservation

> it would be useful to provide some comparative data on the general divergence between the two *Fusarium* species to put this statement into context: what is their molecular distance? what is their overall transcriptomic distance?

page 5

The conserved mid-embryonic period was characterised by a down-regulation of evolutionary young genes and an up-regulation of temporally pleiotropic genes

> this statement reads as if active regulation events and pleiotropic functions were traced, but only passive expression patterns are analysed. The more appropriate wording would therefore be (here and in other sections of the manuscript): "The conserved mid-embryonic period was characterised by a lower expression of evolutionary young genes and a higher expression of evolutionary older and therefore more generally expressed genes..." (or similar)

Note also that the correlation between age of the genes and broadness of their expression is well established (mostly due to the large fraction of house keeping genes among the old genes) - one should therefore be careful with interpreting the latter ones as "pleiotropic" without any further distinction

page 11

We reason that constraints on morphology are imposed continuously by regulatory complexity that results from dynamic and tissue-specific transcriptional controls, resulting in a conserved transcriptome. Conversely, unicellular stages are more permissive to evolutionary change.

> these conclusions could also be revisited in view of the third model (Darwin's model): what is the environmental exposure of the single-cell stages versus the multicellular stages in *Ectocarpus*? They may be more susceptible to predators (or viruses) and may therefore have evolved more new genes relevant for their defense.

Referee #4 (Remarks to the Author):

Over the past decade, there has been increased attention to conceptualizing the evolution of embryonic development using the rock-solid observation of the morphological hourglass-shaped pattern of divergence across animal embryogenesis. The hourglass pattern has been probed on the gene regulatory level using various comparative transcriptomics technologies often creatively combined with phylogenetic relationships among the genes and/or species under comparison. These studies established the hourglass pattern on the gene regulatory level for animals, plants, and fungi. Lotharukpong et al. extend those studies to a new lineage, the brown algae. Since brown algae evolved multicellularity independently and exhibit a broad range of life cycles that include switching between unicellular and differentially complex multicellular stages, this context provides interesting new insight into the generality and origins of the hourglass patterns.

The authors present three main conclusions. First, they confirm the existence of an hourglass-like pattern of transcriptome utilization across two species of brown algae that undergo a multicellular stage akin to animal development. Second, they corroborate findings in other systems that male gametes present an opportunity for evolutionary young genes to be expressed. Finally, using

transcriptomic analysis of a species of brown alga that alternates between single (gametes or spores) and multicellular stages, they propose that multicellular stages constrain the evolutionary change, confining it to the single-cell stages and the gametes in particular. Overall, this study cements the hourglass model as a useful synthesis concept for evo-devo across the Tree of Life and provides new context and motivation to understand the mechanistic origin of such a apparently universal pattern of genome realization.

Below, I provide three suggestions for improving the manuscript.

1) The establishment of the hourglass pattern in two brown algae with recognizable embryogenesis is convincing. It uses the Transcriptome Age Index (TAI) methodology, which combines gene expression and gene age. While TAI is a complex additive score, it has been used for these purposes several times before. I appreciate the non-parametric permutation testing of the null hypothesis. I wonder, though, to what degree is the hourglass pattern driven by expression versus the gene age? Comparing gene expression across species requires stringent control over the orthology of genes and a reasonable congruence of the progression of developmental stages. I am not an expert on the brown algae in question (or any brown algae ;-)), but the schematics of the developmental stages suggest that the morphology, composition, and timing of developmental events between the two species could be comparable. Therefore, could the authors look for the hourglass pattern using the gene expression comparison alone?

2) The sex-specific differences in TAI indices are convincing. Similar differences could be observed when comparing the "stipe" and, to a lesser degree, "holdfast" parts of the *F. serratus* male to its gamete. Moreover, there is no difference in TAI between "holdfast" and "stipe" and gametes in *F. distichus*. Aren't those parts of the alga body considered vegetative as much as the "veg tips"?

3) The *Ectocarpus* is the most interesting part of the story. The TAI and TDI differences are very significant and interesting (it would be useful to explain the connection between TDI and relaxed purifying selection more explicitly for a potentially broad readership of the paper). I wonder, however, whether the magnitude of this change could mask differences among the three stages of multicellular parts of the life cycle. Could the authors apply the same rigorous statistical testing to the multicellular time course in *Ectocarpus* to explicitly reject the hourglass pattern at this level?

Referee #4 (Remarks on code availability):

They are using well established and vetted open source code packages.

Referee #5 (Remarks to the Author):

The manuscript by Lothakpong et al tests the generality of the developmental hourglass model of embryonic development by examining gene expression profiles at different developmental stages of two distinct genera of the Phaeophyceae (brown algae) belonging to the stramenopiles. Specifically, the work sets out to determine whether an early conservation model or hourglass model applies to these organisms. The manuscript addresses a major fundamental issue in evolutionary and

developmental biology and the outcomes will be of wide interest to both evolutionary and developmental biologists. Studies such as these are essential in order to better understand the principles of developmental pattern and its adaptation to changing evolutionary pressures. It is clearly presented in all aspects would be easy to understand by a wide audience. This is potentially a landmark paper in this field.

While prior work with animal and plant embryos provides support for the developmental hourglass model, the current manuscript presents a number of novel features. It tests more widely the generality of the hourglass model by utilizing a group of organisms that evolved multicellularity quite independently. Moreover, the inclusion of *Ectocarpus* in this study, which does not display canonical embryonic development has allowed the respective influences of multicellular development (*Ectocarpus*) and embryogenesis (*Fucus*) to be neatly teased out. An additional novel feature of this work is that it allows selection in gametes to be identified.

The analyses appear to have been carried out thoroughly, allowing the determination of evolutionary age of expressed genes at different developmental stages and different cell or tissue types (reproductive, vegetative, gametes, spores etc), using appropriate indices (TAI, TSI). The life cycles of *Fucus* and *Ectocarpus* are clearly explained for the reader not familiar with the brown algae. Supplementary information and figures are appropriate and necessary. The results with *Fucus* embryos indicate a transcriptome hourglass pattern, similar to those found in animals and plants and indicate that male reproductive tissue and sperm display an evolutionarily younger transcriptome supporting the generality of the model. The *Ectocarpus* study reveals that multicellularity is associated with a more conserved transcriptome compared with unicellularity. This work also showed quite nicely the lack of sexual differences in transcriptomes in *Ectocarpus*, consistent with the sexual isogamy in this species in contrast to the egg-sperm dimorphism in *Fucus*.

Overall, appropriate statistical analyses appear to have been carried out. However, I would like to have seen more detailed statistics for the differences in TAI of the different tissues in Fig 3. The error bars for male gamete TAI of *F. serratus* are large. Are the male gamete values significantly different from female? Also, are the differences between male and female gametes of *F. disticus* really significant?

Ideally, inclusion another species, such as *Macrocystis* or *Laminaria*, which display both canonical zygotic sporophyte development as well as modular multicellular gametophyte development could potentially allow both embryonic development and multicellularity to be dissected in a single species. However, given the amount of work involved, along with the likely technical difficulties in culturing and isolating these life cycle stages, I consider that this would be beyond the scope of this manuscript.

Referee #5 (Remarks on code availability):

I was able to access the code at https://github.com/LotharukpongJS/hourglass_brownalga. It appears to be sufficiently well documented and described to enable it to be run. However, I was not able to test test the code.

Author Rebuttals to Initial Comments:

We extend our most sincere thanks to the reviewers for their helpful feedback and constructive critiques that helped us significantly improve the manuscript.

Referee #1 (Remarks to the Author):

Lotharukpong et al set out to explore transcriptome conservation patterns in brown algae and specifically test if the developmental hourglass previously described in plants, animals and controversially in fungi is tied to embryogenesis or to complex development in general? This is a great idea and if confirmed could contribute to solving a long-standing debate and explain why the hourglass model has been accepted more in certain lineages (e.g. animals) whereas debated in others, where embryogenesis is missing (fungi). This is a well-written and resource-rich study conducted with high methodological rigor (with some exceptions, see below). The complex analyses are well-presented in an easy-to-follow narrative and the independent evolution of embryogenesis in brown algae offers a good model system.

However, I think the ms currently falls short in testing whether the hourglass is tied to embryogenesis or complex development, due to sampling and analytical issues explained below. If the Authors really want to test this then they should test the presence of an hourglass in embryogenetic (done) and multiple multicellular but not embryogenetic processes. A systematic analysis for the latter is missing, only 1 dataset was analyzed. I think this constrains the generality of the findings. This could be remedied by analyzing more nonembryogenic multicellular developmental processes.

Re: We are grateful for this detailed analysis of our manuscript, and the excellent suggestion to add additional non-embryogenic datasets to strength our narrative. We confirm that our idea is that complex multicellular generations that go through canonical embryogenesis to determine their body pattern do exhibit the typical hourglass molecular pattern, whereas more 'simple' morphologically less complex species or generations presenting modular development and without a canonical embryogenesis present still a developmental transcriptome that is characterised by the expression of older genes in multicellular stages and younger genes in unicellular stages.

To address the reviewer comment, we have included developmental transcriptomes from two additional species: the kelps *Saccorhiza polyschides* and *Laminaria digitata*. These two kelp species have a divergence time of >100 MY, and belong to two different brown algal orders, Tilopteridales and Laminariales. The new datasets contain several non-embryogenic multicellular developmental stages for these species. Moreover, these species display both canonical embryogenic development (in the sporophyte generation) as well as modular multicellular development (in the gametophyte generation), which allowed both embryonic development and multicellularity to be dissected (see request from **Reviewer 2 and 5**).

As the reviewer will hopefully appreciate, the inclusion of these new datasets has significantly strengthened our findings. The multicellular stages that could be sampled clearly show that the findings of *Ectocarpus* can be extended to the modular gametophyte stages of other brown algae clades. Moreover, although we miss the very first stages of embryogenesis in *Laminaria digitata*, the resolution is sufficient to capture the 'second part' of the hourglass, with transcriptome conservation patterns decreasing once the transition to proliferation stages occurs.

In conclusion, the new datasets and analysis supports the hypothesis that the developmental hourglass is tied to embryogenesis in complex multicellular generations in brown algae, and that the transcriptome is most conserved in multicellular stages, compared to a unicellular stage, in the generations or species that show modular development.

We would like to emphasise that due to the specificities of the biological material, these samples were extremely challenging to obtain, and unfortunately it was impossible to obtain complete datasets for 'all'

stages across all the life cycle for these extra species. For example, the germination and first cell division stages of the embryo development in the kelp *L. digitata* simply cannot be obtained, because the fertilized egg stays attached to the maternal gametophyte which makes it basically impossible for us to guarantee that no maternal tissue is present. These technical limitations are mentioned in three sections of the manuscript:

Line 417-420: “Meanwhile, in *L. digitata* and *S. polyschides*, access to biological material was extremely challenging, resulting in a less complete dataset for these species. For example, the very early sporophyte stages of *L. digitata* were inaccessible due to contamination from the maternal gametophyte.”.

Line 972-973 Figure S12 . “Note that earlier sporophyte embryogenesis stages could not be sampled due to potential contamination from maternal gametophytic tissue.”

Line 115-119 - “In order to further broaden the species diversity, we analysed transcriptomic data from a subset of developmental stages in two different kelp species *Laminaria digitata* and *Saccorhiza polyschides*. Both species alternate between filamentous gametophytes (which mirrors *Ectocarpus* ‘simple’ development) and more complex sporophytes (which mirrors *Fucus* development).”

Due to the above mentioned caveat, it is not clear how the Authors arrived to the conclusion Abstract““Our findings suggest that transcriptome conservation in brown algae is associated with cell differentiation stages, but not necessarily linked to embryogenesis.”. If anything, it looks like it is linked to embryogenesis, though clarity on whether a phase of cell differentiation exists in the comparator species (*Ectocarpus*) and the apparent lack of an hourglass in this taxon makes this conclusion confusing.

Re: We understand that we were perhaps not sufficiently clear in our narrative. What we meant is that the level of transcriptome conservation during development may not be strictly linked to embryogenic processes *per se*. In the case of organisms that go through modular development (where any cell of the adult algae may differentiate into any other cell type), there is not a precise stage of their development where body plans are settled (they are ‘continuously reiterated’) - we believe this feature blurs the detection of changes at the resolution of the whole organism, and leads to the observation that multicellular stages, as a whole, have more conserved transcriptomes (expressing older genes) as opposed to unicellular stages that are more rapidly evolving (expressing younger genes). In other words, we believe that a classical “hourglass pattern” is only present in more complex brown algae (such as *Fucus*) during embryogenesis. In *Ectocarpus* and in gametophyte stages of kelps, which display modular development, we see only a younger TAI in unicellular versus multicellular stages. We note that excluding the unicellular stages and focusing the analysis only on the multicellular stages as suggested by **Reviewer 2** actually demonstrates there is *not* a consistent hourglass pattern in multicellular development of species with modular growth.

The manuscript has been revised to clarify these ideas in the light of the newly included datasets.

Results - it is great and important information that Fucales species can be sampled without contamination from parental tissues. What is the situation in *Ectocarpus*? Given the importance of this species in the comparisons it may be important to note how this species can be sampled.

Re: It is indeed crucial to ensure that no parental tissue is contaminating the samples. This was actually an important feature to take into consideration in our choice of the species of brown algae we analysed. For example, unlike *Fucus*, kelps’ early zygotic development is virtually impossible to collect because zygotes remain attached to the maternal gametophyte – so exhaustively assessing hourglass patterns in embryogenesis in species other than *Fucus* is very difficult. Note that this is the reason why we could only obtain fragmentary stages for the two additional kelp species presented in the revised manuscript.

We confirm that in *Ectocarpus* all life cycle stages can be easily obtained without contamination of parental tissues. Note that it was for this reason we chose to work with haploid sporophytes of *Ectocarpus*

as representatives of the sporophyte generation in *Ectocarpus*, as they faithfully reproduce the developmental patterns of a diploid sporophyte (Coelho et al, PNAS 2011; Godfroy et al, Development 2023).

Results/“using tau as a proxy for pleiotropy” - I challenge using tau as a proxy. Tau measures to what extent a gene is ‘housekeeping’ or ‘tissue/stage-specific’. Pleiotropy refers to alternative (not constantly needed/rarely needed) functions of a given gene. I doubt tau can substitute more sophisticated measures of pleiotropy available .e.g in yeast based on experimental data. Therefore, I agree with the conclusion “were represented by more broadly expressed genes (i.e., low tau) whereas early and later developmental stages were characterised by more stage-specific genes”, but not with the next sentence (Therefore, more pleiotropic genes are associated...). The references provided do not support the usage of tau as a substitute of pleiotropy.

Re: Indeed, the relationship between *tau* and pleiotropy is based on the idea that genes expressed in many tissues are more likely to affect multiple traits than genes expressed in fewer tissues. Hence, we used *tau* as a proxy of the pleiotropic effect of a gene. Note that *breath of expression* has been shown to correlate with QTL-based pleiotropy measures (Watanabe et al, Nature Genetics 2019) and is an established estimate of pleiotropy (e.g. Thorhólludottir et al, Evolution 2023). However, we take the reviewer’s point and we now modified the sentence to (Line 165-168). If the reviewer still believes that the pleiotropy term should not be used at all, we can of course delete it.

Results/Fucus TAI profiles - the methods mentions the flatline and reductive hourglass tests for assessing the significance of the hourglass shape. I think should be mentioned here, in the main text what p-values did these tests give for Fucus.

Re: Thank you for pointing this out. In the main text when introducing the Fucus TAI results, we now write: “The TAI profiles were robust to all RNA-seq data transformations, consistently returning significant p-values (< 0.05) for both the flat line and the reductive hourglass tests (Methods; Fig. S2).” (Line 138-140) and clarified in the methods section that “The p-value defines (for each tested shape) the probability that the observed TAI pattern is drawn from a random set of TAI profiles with permuted gene ages.” (Line 508-510).

Results/Fucus TAI profiles - An uncertainty here is that in *F. distichus* the ‘waist’ of the hourglass is stage 4.5, which is not defined anywhere in the text (in particular, missing from Figure S5 defining the stages). What does this stage correspond to, was it obtained by some extrapolation from stages 4 and 5?

Re: We apologise for having missed this in **Fig. S5**. To clarify, the data was *not* obtained by extrapolation from stages 4 and 5. We have now included stage 4.5 in *F. distichus*, which is the stage 4 and 5 seen in *F. serratus*. We note, as shown in **Fig. S4**, that there is a “shift in the timing of the developmental stages between the two species” (most conserved stage S4 in *F. serratus* is closest in transcriptome composition to stage S4.5 in *F. distichus*). We modified the results for clarity: “We note a shift in the timing of the developmental stages between the two species, where the transcriptome of the *F. distichus* stage with the lowest TAI (S4.5) was most similar to the *F. serratus* stage with the lowest TAI (S4) (Fig. S4).” (Line 143-145).

Throughout the ms the term “evolutionary transcriptomes” seems superfluous. The Authors talk about transcriptomes, I don’t see how a transcriptome could be ‘evolutionary’. Please consider rephrasing.

Re: We used the term “evolutionary transcriptome” to describe a set of evolutionary methods (e.g., TAI) applied to the transcriptomic analysis. For example, in the Results subsection “Distinct evolutionary transcriptomic patterns mark *Fucus* adult tissues and sexes”, we believe it is informative to describe the

differences in the *evolutionary* transcriptome rather than transcriptome in general, since without the evolutionary aspect of the finding, we cannot directly relate this to the “out of male receptacle” hypothesis. However, where it is deemed superfluous, we have replaced the term “evolutionary transcriptomes” to “transcriptomes”, “developmental transcriptome profile” or “evolutionary transcriptomic patterns” for clarity. If the reviewer still believes that “evolutionary” should not be used at all, we can remove it.

Throughout the text, please provide exact p-values, not cutoffs (e.g. <0.05).

Re: We apologise for not having been exact enough here. Note that probabilities <2e-16 are not exactly computed in R, so we now show all exact p-values above this threshold and write <2e-16 otherwise.

“In *F. serratus* males, different tissues presented significantly different TAI values (flat line test; $p < 0.001$)” - The flat line test might not be an appropriate test here. The Authors are basically comparing age distributions of expression weighted gene ages. I suggest comparing transcriptomes in a pairwise manner. The flatline test can be misleading if only a single transcriptome differs from the rest. An example to this: “This difference was also observed in *F. distichus* (Fig. 3C)” - Based on Fig3C only vegtip seems to differ, all others look the same. Also, could vegetative and reproductive tissues be clearly marked?

Re: This is an excellent suggestion and we thank the reviewer for proposing the “pairwise TAI test”, which we now implemented into myTAI (as the PairwiseTest() function) for anyone to use and which we now employ for comparing the transcriptomes in a pairwise manner. In detail, the PairwiseTest () performs a permutation test to assess the difference between the mean TAI values between specified contrast. The corresponding p-value quantifies the probability that a given TAI pattern (or any evolutionary transcriptomic pattern) does not differ from the alternative hypothesis (specifying the direction of difference). A p-value < 0.05 indicates that the corresponding phylotranscriptomics pattern does indeed differ. We have now added the respective p-values based on the pairwise organ/sex analysis to the results section: “*In F. serratus* males, TAI values differed significantly between tissues (flat line test; $p = 8.57 \times 10^{-4}$), with reproductive tissues (reproductive tip) exhibiting a markedly higher TAI compared to vegetative tissues (vegetative tip, holdfast and stipe) (pairwise TAI test; $p = 0.00534$) (Fig. 3B). This difference was also observed in *F. distichus* (pairwise TAI test; $p = 0.00240$) (Fig. 3C). In contrast, in *F. serratus* females, TAI values did not differ significantly between vegetative and reproductive tissues (pairwise TAI test; $p = 0.407$) (Fig. 3B). We also noticed that sexual differences of TAI were more pronounced at the reproductive tip in *F. serratus* adults, with males having a markedly younger transcriptome compared to females (pairwise TAI test; $p = 2.59 \times 10^{-6}$).” (Line 198-206)

“Multicellularity constrains transcriptome evolution in *Ectocarpus*” - My concern with this chapter is that it does not explicitly look for an hourglass, as it should, based on the aim stated. The chapter shows that multicellular stage transcriptomes are more conserved than those of unicellular stages. How does this result relate to an hourglass? Is there an hourglass during gametophyte and/or partheno-sporophyte development?

Re: **Reviewer 1** raises a very interesting point and debate. Whether this pattern of ‘multicellularity constraint’ also follows an hourglass pattern itself or whether the constraint acts purely on the transition from a ‘unicellular cell fate identity’ to a ‘multicellular cell fate identity’ and the observed hourglass pattern found in embryo development is then a downstream consequence of this transition constraint (hypothesized under the term ‘organizational checkpoint’ in Drost et al., 2017 Curr Op in Gen & Dev) remains to be explored and tested. Since our newly added datasets suggest a ‘multicellularity constraint’ during the modular growth in the gametophyte generation, we posit that the hourglass pattern itself could originate from the establishment of multicellularity and we hope that the community will pick up on this point and collect further evidence in this direction. We raised this point in the discussion: “We posit that

these constraints may have become concentrated towards a more singular 'mid-embryonic period' during the evolution of lineages with increased morphological complexity such as *Fucus*." (Line 371-373)

When we analysed the *Ectocarpus* multicellular stages *only*, we did not find an hourglass pattern except in male partheno-sporophyte. Due to inconsistencies with other multicellular stages, we take a more conservative interpretation that multicellularity stages have a lower TAI *in general* rather than a distinct hourglass pattern. This has been described in the result: "When restricting the reductive hourglass test to multicellular gametophytic or partheno-sporophytic development, only the male partheno-sporophytes returned a significant hourglass shape (reductive hourglass test; $p = 4.22 \times 10^{-15}$). Since this pattern is not consistent in other instances of multicellular development (male gametophyte, female gametophyte and partheno-sporophyte) (Table S4), we interpret that multicellular stages in these filamentous brown algae have a lower TAI overall, rather than displaying a canonical transcriptomic hourglass." (Line 238-244)

Why are unicellular stages involved in the analysis? Sperms and eggs were not involved in tests of animal/plant hourglasses either, so it feels the analyses should be restricted to stages of the multicellular structure. Or should these analyses be not viewed as an attempt to find an hourglass? Maybe there was an hourglass if only multicellular stages are considered?

Re: **Reviewer 1** is absolutely correct that both animal and plant transcriptomic hourglass studies did not involve sperm and egg cells for their analysis. As the reviewer correctly assumes, we chose to include them in our study for *Ectocarpus* because, unlike animal/plant studies, these unicellular stages can generate a new multicellular individual even in absence of gamete fusion. Thus, in these more simple brown algae, unicellular stages play a developmentally similar role to zygotes in animal and plant studies. However, we also tested the hourglass pattern in multicellular stages *only* and this has now been described in the revised manuscript. Please see the changes to the manuscript in the response above for further clarity.

In *Fucus*, "conservation was coupled to the transition from cell differentiation to proliferation". Are there homologous/analogous steps during multicellular development of *Ectocarpus*? What happens before the establishment of the immature gametophyte stage? Is there a stage, like early colony formation, focusing on differentiation, followed by largely just growth? Could those stages be sampled?

Re: We thank the reviewer for raising these points. We confirm that there is no analogous developmental scenario in *Ectocarpus*, and in any other more "simple" brown algae and it should also be noted that multicellularity in brown algae is clonal rather than aggregative. Although the first cell division of the gametophyte leads to a rhizoid and an upright cell in *Ectocarpus*, eventually upright cells are able to produce more rhizoids. In other words, all cells in these filamentous brown algae have high cell potency (most being 'totipotent' if not 'pluripotent') and growth is highly modular. Note also that the gametophyte generation of kelps is also modular, and accordingly, we find no pattern of hourglass in these generations (new figure **Fig. S12**) but we still find that multicellularity is more constrained than unicellular stages.

In sum, *Ectocarpus* development is not marked by a differentiation stage followed by a growth phase (this is the reason why we state: "[...] transition from cell differentiation to proliferation" [Line 281-292]).

Therefore, unfortunately the experimental design suggested by the reviewer cannot be achieved.

I think the essence of the comparison lies in identifying and testing the developmental period of *Ectocarpus* that may show signs of differentiation and proliferation (maybe the early stages of the establishment of the colony?). The current dataset in my opinion is not going to depths for testing this and uses a suboptimal sample set. It would be important for such a key conclusion to not hinge on incomplete exploration of the sample space.

Re: While we fully understand the rationale of **Reviewer 1**, as we pointed out in the previous discussion, *Ectocarpus* follows a modular growth that is driven by cells with relaxed potency (i.e., analogous to

totipotent cells in animals, which can differentiate into any cell types). Specifically, the round, basal cells in a filament can differentiate at any time during development into other cell types, including upright filaments, reproductive cells, elongated apical cells. We hope that our inclusion of unicellular vs 'modular' multicellular stages from two different kelp species makes a strong case that cell fate changes from unicellularity to multicellularity are marked by younger vs older transcriptomes. We also note that restricting the analysis to only multicellular developmental stages of *Ectocarpus* does not show a canonical embryogenic hourglass pattern.

Fig4 please define abbreviations

Re: Thank you for spotting this. We now define all abbreviations.

Though I'm not a brown alga expert, it looks like *Fucus* and *Ectocarpus* all have gametes and multicellular gametophyte stages.

Re: *Ectocarpus* alternates between gametophyte and sporophyte generations. However, *Fucus* does not have a gametophyte generation – it has a diploid life cycle, analogous to that of an animal. Both have gametes.

That is, a comparison like Fig 4 could be performed in *Fucus* also and it could be tested how the pattern there relate to the hourglass in embryogenesis. This would contribute to the more thorough testing I miss for substantiating the main conclusion.

Re: While this is an excellent suggestion, unfortunately *Fucus* does not have a gametophyte generation, so we cannot design an experiment in this direction. However, our inclusion of the developmental transcriptomes from two additional kelp species (which have modular gametophyte and complex sporophyte generations) addresses the reviewers' points and provides further evidence that (simple modular) multicellular stages express older genes than unicellular stages.

Discussion/"pointing to the redeployment of pre-existing molecular toolkit for brown algal complex multicellularity" - this is a fashionable and abstract idea these days, but in this specific context, with a stramenopile ancestry, what pre-existing toolkit should be envisioned to that could underpin the body plan? Isn't it possible that simply changes in the expression regulation of conserved genes represent the innovation underlying multicellularity? This scenario is less abstract, but offers an easier to grasp explanation, that is, simply innovation but not in gene content (as is predicted by the cis-regulatory hypothesis).

Re: We agree with Reviewer 1's assessment, and simply followed the proposed toolkit model to communicate the abstraction of innovation through rewiring of expression regulation based on the cis-regulatory hypothesis. Since much more community effort is required in the future to determine what upstream factors led to the expression changes, we now rephrase our statement to: "This pattern, together with data from *Laminaria sporophytes*, suggests that ancient genes (rather than genes specific to brown algae) form the *Fucus* and *Laminaria* body plans, pointing to a cis-regulatory hypothesis for the redeployment of pre-existing genes in the evolutionary innovations associated with brown algal complex multicellularity (Carroll 2008)." (Line 312-316).

Discussion/"from a cell-type differentiation stage where the body plan of the adult *Fucus* is established, to a proliferative stage" - this goes back to my arguments on sampling. If this is the case, then an effort should be made to identify differentiation to proliferation switches in *Ectocarpus* and the hourglass should be tested there.

Re: We understand the reviewer's idea, but as we discussed above due to the lack of "differentiation-to-proliferation" stages that can be sampled from *Ectocarpus*, using *Ectocarpus* as a model system for this question is (unfortunately) biologically not feasible. We hope the reviewer agrees that the inclusion of the two additional species strongly reinforces the ideas we raise in the manuscript.

Discussion/"We reason that constraints on morphology are imposed continuously by regulatory complexity that results from dynamic and tissue-specific transcriptional controls, resulting in a conserved transcriptome." - please check sentence, too abstract and/or makes little sense

Re: We now clarify this sentence by rephrasing to: "*We reason that the regulatory complexity from cell-type differentiation programmes activated across filamentous multicellular stages results in an overall conserved transcriptome.*" (Line 369-371)

Discussion/"this pattern may represent a fundamental characteristic of multicellular complex development across the entire eukaryotic tree of life" - This conclusion seems to be in conflict with the finding that there was an hourglass in *Fucus* embryogenesis but not in *Ectocarpus*. If this conclusion holds then why does *Ectocarpus* not have an hourglass? I accept that the presence of the hourglass in other taxa partially supports this, but then its not a conclusion drawn from these data. It should be noted though that in previous studies a clear distinction between complex development and embryogenesis was not made - the key good idea in this paper.

Re: What we meant here was that embryogenic (molecular) hourglass indeed is a common feature of multicellular organisms with a higher degree of complexity (in development and morphology). This level of complexity found in *Fucus* contrasts with the simpler filamentous form of (modular) multicellularity found in *Ectocarpus*, which exhibits a conserved transcriptome during multicellular, more complex, stages.

We now rephrase to: "*Our distinction between complex multicellular development and embryogenesis suggests that transcriptome conservation patterns are a fundamental characteristic of complex multicellularity itself, with possible downstream effects on embryogenesis.*" (Line 386-389) to make this distinction clearer.

Methods/Distance/similarity-based transcriptome comparison - here, the Authors make expression comparisons across species. They primarily use TPM for quantifying expression, which is not normalized for unit gene/transcript length. As a result, if 1-1 orthologous genes have length differences between the compared species, this analysis will provide misleading results. The Authors should use length-normalized expression metrics (e.g. FPKM) or demonstrate that gene length differences are not affecting their results.

Re: We used gene-length normalized TPMs for the analysis and have now clarified the point in the Methods: "Note, expression was quantified using length-normalised TPM to avoid bias from different gene lengths between species." (Line 572-573)

Methods/enrichment - how was 'contributing most' defined?

Re: 'Contributing most' referred to "GO enrichment analysis was performed on genes contributing most to *Ectocarpus* TAI (inferred from the partial TAI value of each individual gene, $pTAI_i$)". We now clarified this in the manuscript (Line 585-586).

In supplementary excel files columns and abbreviations are not explained. Legends should be included.

Re: We now explain all abbreviations in the supplementary excel files and added legends to clarify the presented data.

Referee #2 (Remarks to the Author):

Review of "A Transcriptomic Hourglass in Brown Algae" by Lotharukpong et al.

In this paper, the authors show that the developmental hourglass model, previously established in animals, plants and fungi, also extends to brown algae- another clade of complex multicellularity. This is a novel and significant paper with important implications for our understanding of the evolution of multicellular development. From what I can tell, the data and analyses are sound, and the paper is well written. My main concern is that it is effectively two phylogenetically independent data points (one simple, and one complex multicellular brown algae lineage), which may not be robust.

Re: We are grateful for the positive feedback from Reviewer 2 and their excellent suggestions. We have now added two more developmental transcriptomes from two different kelp species (see also response to Reviewer 1). The inclusion of these two additional species substantially reinforces our conclusions and we thank again the reviewers for having made this important suggestion.

Summary of key results: First, the authors demonstrate that two species of complex multicellular brown algae, *Fucus serratus* and *Fucus distichus*, exhibit a transcriptomic hourglass pattern during development. The 'waist' of the hourglass, representing the stage with the greatest conservation in the transcriptome, is mainly expressing old genes (thing that pre-date the evolution of brown algae themselves), and the repression of young genes. The waist of the hourglass corresponds to the portion of time in which the embryo is defining its body plan, which makes sense, as this is arguably the most critical stage in development that is under the greatest evolutionary constraint.

Second, the authors show that in the filamentous brown alga *Ectocarpus*, which lacks a embryogenesis and instead grows in a modular manner, the transcriptome is most conserved in multicellular stages, compared to a unicellular stage. This shows that transcriptional constraints are not simply a feature of embryogenesis, but the life history stage in which morphological patterning is being determined.

Finally, the authors show that male reproductive tissues are enriched in young genes in *Fucus*, mirroring the "out of testis" pattern seen in animals and plants. I'm less sure what to make of these results, as I am less familiar with the literature on the evolution of sex-specific tissues.

The developmental hourglass model has been intensely studied in animals, with some recent work in plants and fungi. This is the first test of the model in the brown algae, which independently evolved complex multicellularity. The finding that brown algae also exhibit a developmental hourglass is significant, as it suggests this may be a universal feature of complex multicellularity and the constraints that come from the developmental phase in which 3D body plans are defined. By comparing species with and without embryogenesis, the authors further provide new insight into the relative contributions of multicellularity and embryonic development to the hourglass pattern. I think that this work will be of broad interest to the evo-devo / major transition communities.

In terms of methodology, the authors generated high quality stage-specific RNA-seq data for three diverse brown algal species. They employed standard comparative genomic methods to infer the evolutionary age of each gene and computed standard metrics (TAI, TDI, TSI) to quantify transcriptome conservation across development. The statistical analyses, including permutation tests and bootstrapping, seem appropriate to me, but this is getting a bit afield of my expertise.

I don't see any major flaws in this paper, though it does feel a bit small (i.e., data light) for a contemporary Nature paper. I think the largest weakness is that just two species represent complex multicellularity in the brown algae, and they are both members of the same genus. In a sense, they're not independent, as they recently shared a common ancestor (not sure about exact divergence time, but they're clearly closely related as the authors had to exclude them as being too similar: "Fucus species were precluded from this analysis due to the short divergence time between *F. serratus* and *F. distichus*, which meant that more than 10% of genes having dNdS of 0". So for all intents and purposes, there is one phylogenetically independent datapoint with *Fucus*, and one with *Ecocarpus*. I think that seeing a similar pattern in divergent species of complex brown algae (maybe a *Macrocystis*?) would lend a great deal of support to the broader claims. However, I'm not sure if this is feasible, as I do not work with these organisms. This is

not a need to have, but I'd love to hear the authors thoughts on the feasibility of broadening the diversity of their sample.

Re: We greatly appreciate this suggestion and we now added developmental transcriptomes for two different kelp species, *Saccorhiza polyschides* and *Laminaria digitata* to strengthen our claims. *S. polyschides* is a Tilepteridales and *L. digitata* belongs to the Laminariales order. Note that we did our best to collect as many stages as possible, but due to technical and biological constraints it is not possible to obtain a collection of stages that is as exhaustive as those for *Ectocarpus* and *Fucus*. We hope the reviewer agrees that the stages of development we obtained are appropriate to further substantiate all our claims. The technical limitations inherent to these organisms have been noted in the methods, **Fig. S12** and the results (see reply to Reviewer 1; Line 115-119, Line 417-420, Line 972-973).

Another suggestion (that is not a requirement!) is that recent single-cell RNA-seq studies in animals have shown that the waist of the hourglass corresponds to the emergence of phylum-specific cell type transcriptional programs (i.e., see 10.1126/science.aar5780 and 10.1038/s41586-019-0933-9). Do we see anything similar here? I'd love to see an elaboration of the intersection of the gene families coordinating multicellular development and the temporal series of development.

Re: This is indeed an excellent suggestion and while the proposed single-cell RNAseq studies still require more technical advancement in the brown algal system due to low-input constraints, we will hopefully revisit this idea once our molecular technology has caught up. Regarding the intersection of gene families coordinating multicellular development and the temporal series of development this is also an interesting idea. Unfortunately, functional genomics analysis in brown algae still lags behind that of animal and plant models organisms, therefore it is very challenging to have a thorough GO term analysis and a clear idea of gene function, especially in the *Fucus* systems. More than half of the genome of these organisms correspond actually to "unknown protein". In an attempt to address the reviewer's suggestion, we have examined if transcription factor families are associated with particular timings of development. As seen in **Reviewer Fig 1** (below), we note that the approx. 200 putative TFs (inferred from GO analysis) in *Fucus* are lowly expressed compared to all other genes (A & B). Moreover, only a handful of these are differentially expressed across development (inferred using the likelihood-ratio test [LRT] from DESeq2 package) (C & D), suggesting that more developmentally significant TFs can still be found and the currently identified putative TFs may not be associated with development. These two factors, combined with the limitation in accurately extracting TFs from homology (especially fast-evolving and/or evolutionary novel TFs), likely results in the lack of TF-enrichment at any particular developmental stage. This unfortunately rules out any *sensitive* analysis of gene families coordinating multicellular development and the temporal series of development. For this reason, we would prefer to preclude this analysis from our study (but of course are happy to include them if the reviewer thinks this is an important aspect). Advances in developmental genetics and molecular biology tools in brown algae will bring us closer in understanding the role of phylum-specific cell type transcriptional programs in forming body plans in

complex multicellular brown algae, and hopefully this type of analysis can be performed more confidently in the future.

Reviewer Figure 1. Expression dynamics of putative transcription factors (TFs) inferred from GO analysis in *Fucus*. Mean expression level (TPM) of putative TFs in relation to the rest of the genome in (A) *F. serratus* and (B) *F. distichus*. Both axes were sqrt-scaled for visualisation. The number of putative TFs differentially expressed across embryo stages (using likelihood ratio test; LRT) in (C) *F. serratus* and (D) *F. distichus*. Expression level ($\log_2(\text{TPM} + 1)$) of putative TFs across embryo stages in (E) *F. serratus* and (F) *F. distichus*.

Finally, this is not a suggestion for this paper, but for future work. There's growing evidence that evolution of non-coding cis-regulatory elements plays a key role in developmental evolution, often by modulating the expression of developmentally-important genes. Specifically, cis-regulatory changes in non-coding elements have been shown to modulate developmental genes and contribute to morphological evolution across various animal taxa. If, for example, developmental enhancers are slow evolving (as has been shown in *Drosophila*), then this might contribute to the hourglass pattern in brown algae. You could figure this out by ChIP-seq or ATAC-seq.

Re: We appreciate this suggestion and we agree that such analysis is out of the scope of the current manuscript. We hope, however, that we can incorporate ChIP-Seq and ATAC-Seq experiments in future work. Our laboratory is currently developing ATAC-Seq protocols for brown algae so indeed these suggestions are in line with our endeavors. Regarding cis-regulatory changes, we raised its potential role in modulating the expression of genes pre-dating the origin of brown algae during the mid-embryonic period in *Fucus*: Line 312-316.

I found the paper relatively easy to read, once I had memorized what TAI, TDI, TSI were. I appreciated that the authors explained their conceptual meaning in the figure legends, and would appreciate even a bit more of this if possible. I thought the overall conceptually-driven approach to the writing was quite effective and clear.

Re: In accordance with some points that were also not clear to the other Reviewers, we further clarified

some of our conceptual meaning and added more details to the relevant figure legends, such as **Fig. 4** (explaining the abbreviation).

Referee #2 (Remarks on code availability):

I did not review the code. I've never run this kind of bioinformatics analysis, so I'm not the right person to review it.

Referee #3 (Remarks to the Author):

Lotharukpong et al. present an excellent dataset on the topic of transcriptomic hourglass patterns in brown algae. Their overall findings are very similar to what was shown before for animals and plants, at least for two of their study species (*Fucus*). The most exciting part of these findings is that the multicellular life cycles of brown algae evolved independently of those of plants and animals. This implies that there is a generality in these patterns, which establish them as one of the few general biological rules.

Given this very general finding, it would be great if the authors would discuss more the actual genes that are causing this average pattern. There is some discussion of "pleiotropic genes", but this definition is too general, since it seems to include both house-keeping genes, as well as ancient regulatory genes or genes required for cell-cell communication. It would be great if the authors could add some additional analyses on the subset of genes that are thought to have arisen in the context of animal/plant multicellularity. Are they the same that are also enriched in the "waist" of the *Fucus* hourglass stages?

Re: We thank Reviewer 3 for taking the time to carefully assess our manuscript and for the excellent suggestions to improve our manuscript.

Note that the definition of pleiotropic genes has been refined in response to Reviewer 1.

In terms of the sets of genes that are related to specific developmental stages (such as the ones at the waist of the hourglass), we find that genes that arose in the context of complex multicellularity in brown algae are *not* enriched in the waist of the *Fucus* hourglass (**Fig. 2C and D**). The evolutionary age of these genes is now further clarified in the results section: "*Interestingly, we noticed that evolutionarily young genes (i.e., genes associated with the origin of complex multicellularity in brown algae in phylostratum [PS] 7, and species-specific genes in PS 8) were markedly downregulated during the waist of the hourglass, rather than older genes being upregulated at this stage (Fig. 2C, D).*" (Line 148-151).

We tried to find a subset of genes enriched at the "waist" of the *Fucus* hourglass (such as transcription factors) but unfortunately, we did not find any functionally enriched classes (see response to Reviewer 2). As we mentioned in the response to Reviewer 2, this is likely caused by the lack of functional annotation of these non-model organisms.

In *Ectocarpus*, the question of what genes are causing the average pattern has been explored from the partial TAI analysis (based on the contribution of each gene to the global TAI pattern). We found that the significant contributors to the patterns in unicellular stages did not return any functional enrichment probably because less than 10% of these genes have functional annotation. In multicellular stages, this enrichment analysis returned older genes with GO term enrichment linked to translational, organellar and transcriptional processes. These results are mentioned in Line 262-271.

I list a number of further more detailed comments below that could improve the paper. Given that there are no line numbers, I reproduce their text sections and add my comment after ">"

Introduction

With the rise of complex multicellular lineages, a major question is how developmental processes can accommodate evolutionary change. In other words, how can the benefits of evolutionary change be co-opted without disrupting intricate developmental processes?

> not sure what this statement is alluding to; could be omitted

Re: We agree and have now removed the latter sentence and modified the earlier sentence to accommodate the change in narrative flow: "*Multicellularity evolved multiple times in eukaryotes (Grosberg and Strathmann 2007). While this evolutionary transition often resulted in relatively simple life forms, in some lineages this transition was followed by a series of evolutionary innovations, resulting in so-called 'complex multicellular' organisms, associated with intricate developmental processes (Knoll*

2011; Niklas and Newman 2013). *The emergence of complex multicellularity is thus thought to be a rare event, having occurred independently in animals, fungi, plants, red and brown algae (Knoll 2011). With the rise of complex multicellular lineages, a major question is how developmental processes accommodate evolutionary change.*" (Line 31-38).

Two models have been proposed to explain how conserved developmental processes...

> There is actually a third model, originally proposed by Darwin: the phylotypic stage is most shielded from the influence of the environment and there are therefore fewer triggers for evolutionary change - see explicit discussion of this point in: Artieri, C. G., Haerty, W. & Singh, R. S. Ontogeny and phylogeny: molecular signatures of selection, constraint, and temporal pleiotropy in the development of *Drosophila*. *BMC Biol.* 7, 42 (2009).

Re: We thank the reviewer for making us aware of this work. We now added this reference and raised Darwin's model regarding the difference in the cell structure and behaviour in unicellular stages of *Ectocarpus* compared to multicellular stages, which could open 'selection opportunity', resulting in a younger transcriptome. *"In conjunction with the pattern of 'multicellularity constraint', unicellular dispersal stages may be more permissive to evolutionary change. Differences in the cell structure of unicells, such as the lack of a cell wall, can result in (a)biotic exposure which opens new 'selection opportunities' (Artieri et al. 2009), compared to multicellular stages. For example, it is well known that virus infection occurs in the unicellular stages (gametes and spores) in Ectocarpus (Müller et al. 1990)."* (Line 374-379).

Note that have now changed the word *explain* to *describe*: *"Two models have been proposed to describe how conserved developmental processes"* (Line 45-46). We reason that 'explains' would denote specific causal mechanisms such as Riedl's idea of 'burden' or 'Darwin's model', rather than the *descriptive* model of early conservation and developmental hourglass, which is more inclusive and purely describes the resulting evolutionary profile.

Figure 1b

> please explain the term "matSP"

Re: We now clarify that 'matSP' refers to 'sexually mature sporophyte'.

Figure 1c

> it is not clear which phases are haploid and which are diploid

Re: We now clarify this by adding (n and 2n) to the respective case in the figure.

page5

...robust to all RNA-seq data transformations (Fig. S2)..

> given that there was substantial discussion on RNA-seq data transformation for studying hourglass patterns, it would be good to expand this section with more details. At the minimum, one should find a more extensive description in the legend of Fig. S2 (and not only hidden in the Methods section)

Re: We agree and have described the motivation in more detail in the legend of Fig. S2: *"Previous studies have shown that RNA-seq transformations can affect TAI profiles (Piasecka et al. 2013; Liu and Robinson-Rechavi 2018), motivating this analysis on previously reported and additional transformations: identity ('none'), square-root ('sqrt'), logarithmic ('log2'), as well as non-parametric ranking ('rank') and variance-stabilising ('rlog')."* (Line 904-907).

Note that we have chosen not to display further TAI profiles from different transformations as a new supplementary figure because they essentially look the same and the key bit, which is the significance of the profiles, remains significant. Instead, we have preferred to describe the motivation behind this analysis further in the figure legend. If, however, the reviewer thinks that showing the further TAI profiles

we can of course do this as further supplemental figures. We would also like to point out that we added all transformation computations in our computationally reproducible scripts on GitHub to point technically interested readers to this transformation analysis.

page5

Despite a shift in the timing of the developmental stages between the two species, the transcriptome of the most conserved stage of *F. distichus* was also the one most similar to the *F. serratus* stage with highest degree of conservation

> it would be useful to provide some comparative data on the general divergence between the two *Fusarium* species to put this statement into context: what is their molecular distance? what is their overall transcriptomic distance?

Re: This is an interesting point. Note that there is no data on the molecular distance between *Fucus serratus* and *Fucus distichus*, but based on Canovas et al 2011 (doi: 10.1186/1471-2148-11-371) we can estimate that the two species diverged about 4 MY ago. This has now been noted in the methods section (Line 556).

The transcriptomic distance (based on expression levels of orthogroups) between *Fucus* species varies by stage and measure (e.g., Pearson correlation, Spearman correlation, Manhattan distance, Jensen-Shannon distance; see **Fig. S4**). Thus, we cannot state an absolute value that would be informative in the context of other cross-species comparisons. Interestingly, the mid-embryo stages generally have a more similar transcriptome compared to the very early and late stages. However, due to the inherent issues associated with precise stage-matching between two species with different developmental speed, we leveraged TAI (which is not affected by this issue) as a measure for transcriptome conservation.

page 5

The conserved mid-embryonic period was characterised by a down- regulation of evolutionary young genes and an up-regulation of temporally pleiotropic genes

> this statement reads as if active regulation events and pleiotropic functions were traced, but only passive expression patterns are analysed. The more appropriate wording would therefore be (here and in other sections of the manuscript): " The conserved mid-embryonic period was characterised by a lower expression of evolutionary young genes and a higher expression of evolutionary older and therefor more generally expressed genes..." (or similar)

Re: We have rephrased the respective sentences in the manuscript as suggested: "*The conserved mid-embryonic period is characterised by reduced expression of evolutionarily young genes and a relatively higher expression of more broadly expressed genes*" (Line 178-180), as well as in other cases where the tracing of regulation events and pleiotropic functions is implied.

Note also that the correlation between age of the genes and broadness of their expression is well established (mostly due to the large fraction of house keeping genes among the old genes) - one should therefore be careful with interpreting the latter ones as "pleiotropic" without any further distinction

Re: We agree with this point which was also raised by Reviewer 1 and we now refined the pleiotropy statement.

page 11

We reason that constraints on morphology are imposed continuously by regulatory complexity that results from dynamic and tissue-specific transcriptional controls, resulting in a conserved transcriptome.

Conversely, unicellular stages are more permissive to evolutionary change.

> these conclusions could also be revisited in view of the third model (Darwin's model): what is the environmental exposure of the single-cell stages versus the multicellular stages in *Ectocarpus*? They may be more susceptible to predators (or viruses) and may therefore have evolved more new genes relevant for their defense.

Re: This is a very interesting suggestion. We now added to the manuscript a reference to this model. Please see our response to Reviewer 1 and the invocation of Darwin's model and the virus in the discussion of the unicellular dispersal stages in a previous reply.

Referee #4 (Remarks to the Author):

Over the past decade, there has been increased attention to conceptualizing the evolution of embryonic development using the rock-solid observation of the morphological hourglass-shaped pattern of divergence across animal embryogenesis. The hourglass pattern has been probed on the gene regulatory level using various comparative transcriptomics technologies often creatively combined with phylogenetic relationships among the genes and/or species under comparison. These studies established the hourglass pattern on the gene regulatory level for animals, plants, and fungi. Lotharukpong et al. extend those studies to a new lineage, the brown algae. Since brown algae evolved multicellularity independently and exhibit a broad range of life cycles that include switching between unicellular and differentially complex multicellular stages, this context provides interesting new insight into the generality and origins of the hourglass patterns.

The authors present three main conclusions. First, they confirm the existence of an hourglass-like pattern of transcriptome utilization across two species of brown algae that undergo a multicellular stage akin to animal development. Second, they corroborate findings in other systems that male gametes present an opportunity for evolutionary young genes to be expressed. Finally, using transcriptomic analysis of a species of brown alga that alternates between single (gametes or spores) and multicellular stages, they propose that multicellular stages constrain the evolutionary change, confining it to the single-cell stages and the gametes in particular. Overall, this study cements the hourglass model as a useful synthesis concept for evo-devo across the Tree of Life and provides new context and motivation to understand the mechanistic origin of such a apparently universal pattern of genome realization.

Re: We are grateful to Reviewer 4 for taking the time to examine our study in detail and for the insightful comments that allowed us to further improve the strength and overall narrative of our manuscript.

Below, I provide three suggestions for improving the manuscript.

1) The establishment of the hourglass pattern in two brown algae with recognizable embryogenesis is convincing. It uses the Transcriptome Age Index (TAI) methodology, which combines gene expression and gene age. While TAI is a complex additive score, it has been used for these purposes several times before. I appreciate the non-parametric permutation testing of the null hypothesis. I wonder, though, to what degree is the hourglass pattern driven by expression versus the gene age? Comparing gene expression across species requires stringent control over the orthology of genes and a reasonable congruence of the progression of developmental stages. I am not an expert on the brown algae in question (or any brown algae ;-)), but the schematics of the developmental stages suggest that the morphology, composition, and timing of developmental events between the two species could be comparable. Therefore, could the authors look for the hourglass pattern using the gene expression comparison alone?

Re: This is an excellent suggestion and we agree that both approaches TAI and comparative transcriptomics can hint toward complementary perspectives of embryonic conservation. In **Fig. S4**, (based on orthogroup analysis, which includes in-paralogues), we have shown that between *Fucus* species, there is an overall tendency to see transcriptome *expression* conservation matching the waist. In the **new Fig. S12**, we find that the most conserved stage in *L. digitata* is closest in transcriptome distance with the most conserved stage in *Fucus*, which suggests that the hourglass pattern can be recapitulated in expression distance. To provide transcriptome *expression* divergence support to the multicellular constraint in *Ectocarpus*, we now provide **Fig. S8** which shows that multicellular stages in *Ectocarpus* tend to be more similar to embryo stages of both *Fucus* species, compared to unicellular stages. Therefore, these analyses show that indeed there is transcriptome expression also conserved during stages with low TAI. The revised manuscript addresses these aspects (Line 245-247).

2) The sex-specific differences in TAI indices are convincing. Similar differences could be observed when

comparing the "stipe" and, to a lesser degree, "holdfast" parts of the *F. serratus* male to its gamete. Moreover, there is no difference in TAI between "holdfast" and "stipe" and gametes in *F. distichus*. Aren't those parts of the alga body considered vegetative as much as the "veg tips"?

Re: Indeed, stipe and holdfast structures can be considered 'vegetative' structures in brown algae. We currently do not have a clear explanation for why male stipe has a higher transcriptome age compared to female stipes, but we can speculate this could be related to the fact that in brown algae the male developmental program is thought to be superimposed on a 'ground state' female development program. It is therefore possible that the female developmental program, even in vegetative tissues, is more 'conserved/pleiotropic compared with the male developmental program. Note that there is currently no detailed information about sexual dimorphism in *Fucus serratus* at the morphological level, and sex biased gene expression has focused on the reproductive structures rather than non-reproductive structures. In other words, though under-explored at the morphological and molecular level, it is very possible that non-reproductive tissues are different between males and females.

We have modified a sentence to address this observation, which we would prefer not to overly speculate on: "*Furthermore, the expression of younger genes in Fucus males is consistent with recent findings in brown algae, implicating the female development program as the morphogenetic "ground state", superimposed upon in males (Cossard et al. 2022; Liesner et al. 2024), though the mechanism behind this pattern in non-reproductive tissues is unclear.*" (Line 346-349)

3) The *Ectocarpus* is the most interesting part of the story. The TAI and TDI differences are very significant and interesting (it would be useful to explain the connection between TDI and relaxed purifying selection more explicitly for a potentially broad readership of the paper).

Re: We agree and now add the following sentence to make this connection between TDI and relaxed purifying selection more clearly in the legend of **Fig 4**: "*Note, the TDI captures the degree of purifying selection because it is based on the dNdS ratio of one-to-one orthologous genes between Ectocarpus sp. 7 and Ectocarpus subulatus, of which >99% is lower than one. Thus, a lower TDI marks a transcriptome composed of genes under stronger purifying selection and vice versa.*" (Line 287-290)

I wonder, however, whether the magnitude of this change could mask differences among the three stages of multicellular parts of the life cycle. Could the authors apply the same rigorous statistical testing to the multicellular time course in *Ectocarpus* to explicitly reject the hourglass pattern at this level?

Re: Absolutely. We now also performed statistical testing for all multicellular stages and also added developmental transcriptomes for two additional kelp species (see response to other reviewers) to further strengthen our overall claims. This has been addressed to Reviewer 1 as well in Line 238-244.

Referee #4 (Remarks on code availability):

They are using well established and vetted open source code packages.

Referee #5 (Remarks to the Author):

The manuscript by Lothakpong et al tests the generality of the developmental hourglass model of embryonic development by examining gene expression profiles at different developmental stages of two distinct genera of the Phaeophyceae (brown algae) belonging to the stramenopiles. Specifically, the work sets out to determine whether an early conservation model or hourglass model applies to these organisms. The manuscript addresses a major fundamental issue in evolutionary and developmental biology and the outcomes will be of wide interest to both evolutionary and developmental biologists. Studies such as these are essential in order to better understand the principles of developmental pattern and its adaptation to changing evolutionary pressures. It is clearly presented in all aspects would be easy to understand by a wide audience. This is potentially a landmark paper in this field.

Re: We appreciate Reviewer 5's analysis and endorsement of our work. The constructive suggestions allowed us to substantially improve our manuscript.

While prior work with animal and plant embryos provides support for the developmental hourglass model, the current manuscript presents a number of novel features. It tests more widely the generality of the hourglass model by utilizing a group of organisms that evolved multicellularity quite independently. Moreover, the inclusion of *Ectocarpus* in this study, which does not display canonical embryonic development has allowed the respective influences of multicellular development (*Ectocarpus*) and embryogenesis (*Fucus*) to be neatly teased out. An additional novel feature of this work is that it allows selection in gametes to be identified.

The analyses appear to have been carried out thoroughly, allowing the determination of evolutionary age of expressed genes at different developmental stages and different cell or tissue types (reproductive, vegetative, gametes, spores etc), using appropriate indices (TAI, TSI). The life cycles of *Fucus* and *Ectocarpus* are clearly explained for the reader not familiar with the brown algae. Supplementary information and figures are appropriate and necessary. The results with *Fucus* embryos indicate a transcriptome hourglass pattern, similar to those found in animals and plants and indicate that male reproductive tissue and sperm display an evolutionarily younger transcriptome supporting the generality of the model. The *Ectocarpus* study reveals that multicellularity is associated with a more conserved transcriptome compared with unicellularity. This work also showed quite nicely the lack of sexual differences in transcriptomes in *Ectocarpus*, consistent with the sexual isogamy in this species in contrast to the egg-sperm dimorphism in *Fucus*.

Overall, appropriate statistical analyses appear to have been carried out. However, I would like to have seen more detailed statistics for the differences in TAI of the different tissues in Fig 3. The error bars for male gamete TAI of *F. serratus* are large. Are the male gamete values significantly different from female? Also, are the differences between male and female gametes of *F. disticus* really significant?

Re: This issue was also raised by two other reviewers and we apologize that adding such a test has escaped our attention. We now introduced a new pairwise statistical test (based on permutation statistics) named `pairwiseTest()` to the `myTAI` package, which we refer to as 'pairwise TAI test' in the manuscript and performed it on our tissue samples. As a result, we find that (1) in *F. serratus*, the male reproductive tip has a significantly younger transcriptome than male non-reproductive tissues, (2) in *F. distichus*, the reproductive tip has a significantly younger transcriptome than non-reproductive tissues, (3) in *F. serratus*, the difference between the female reproductive tip and the non-reproductive tissues is insignificant, and (4) the male reproductive tip has a younger TAI than female reproductive tip. The text has been modified to report this finding (Line 198-209).

Regarding the gametes, the error bars are standard deviations, and the male TAI is not significantly younger than the females in both species, based on permutation statistics. We assume that the large standard deviation in gametes is caused by expression noise (seen also to some extent in *Ectocarpus* data). However, it is interesting to note that male gametes are younger in TAI compared to female gametes in both *F. serratus* and *F. distichus*. In the discussion, we have toned down the causative role of male gametes (“associated with” rather than “driven by” in “This pattern is likely driven by the presence of male gametes (sperm) in the reproductive organs of male individuals” (Line 339-340)).

Ideally, inclusion another species, such as *Macrocystis* or *Laminaria*, which display both canonical zygotic sporophyte development as well as modular multicellular gametophyte development could potentially allow both embryonic development and multicellularity to be dissected in a single species. However, given the amount of work involved, along with the likely technical difficulties in culturing and isolating these life cycle stages, I consider that this would be beyond the scope of this manuscript.

Re: We now also include the developmental transcriptomes in two additional species, *Saccorhiza polyschides* and *Laminaria digitata* (two phylogenetically different kelps) to address this point. Note that these two species are very distantly related to *Ectocarpus* and *Fucus* (see **Fig. S12a**), which allowed us to address the point of Reviewer 1 & 2 about generalisation and the limited number of species used previously.

As Reviewer 5 correctly noticed, biological limitations did not permit us to sample as thoroughly as for *Ectocarpus* and *Fucus* the early stages of embryogenesis, and it was also very challenging to obtain all stages for the same species. However, in both new species, all stages that could be sampled also follow the TAI trajectory that was found in *Ectocarpus* and *Fucus*:

- Similarly to *Fucus*, we find a hourglass ‘waist’ (conservation of transcriptome) during embryogenesis of the kelp *L. digitata*: corresponding to a major ontogenetic transition, from a ‘cell-type differentiation’ stage, where the algal body plan is established, to a more ‘proliferative’ stage. Later stages of development have less conserved transcriptomes.

- The comparison between unicellular and multicellular stages in the gametophytes of *L. digitata* and *S. polyschides* shows that, similarly to *Ectocarpus*, unicellular stages have less conserved transcriptomes. Therefore, the inclusion of these two other species allowed us to open the breath of study species and further confirm the generalisation of the patterns seen in the three other species. (Fig. S11, S12).

Referee #5 (Remarks on code availability):

I was able to access the code at https://github.com/LotharukpongJS/hourglass_brownalga. It appears to be sufficiently well documented and described to enable it to be run. However, I was not able to test test the code.

Thank you for your constructive comment. Previously, the scripts were designed to be run in order from script 1 to 3. We have modified the code such the data is available for all scripts can be run independently. This has been noted in the markdown file.

Reviewer Reports on the First Revision:

Referees' comments:

Referee #1 (Remarks to the Author):

The ms by Lotharukpong et al has improved and I appreciate the detailed responses of the Authors. I think this paper will be an interesting addition to the already sizable literature on the developmental hourglass hypothesis. I have three more substantial suggestions and a few minor ones, that the Authors may find helpful in further improving their ms.

1. A challenge for the reader is that subheadings don't form a coherent storyline, thereby don't help grasping the main message.

Hourglass-shaped transcriptome conservation profiles during *Fucus* embryo development
Distinct evolutionary transcriptomic patterns mark *Fucus* adult tissues and sexes
Multicellularity constrains transcriptome evolution in *Ectocarpus*

Somehow # 2 and 3 don't reflect on the hourglass part of the story and read as if they were addressing very different questions. This lack of coherence is present in the chapters themselves too. I realize this is a complex story, but having a clearer logic would help readers.

2. I think to a large extent clarity is lost on the phrasing. I think it would be important to be very precise about what the hourglass pattern applies to in brown algae. Embryogenesis, complex development, morphologically complex and complex multicellularity are used interchangeably at several places. This causes confusion and is particularly unfortunate given that one of the goals of the paper is to tease apart these concepts and how they relate to an hourglass.

3. 'Out of receptacle' hypothesis. I have reservations about making a comparison with the out of testis hypothesis based on these data. There is no doubt male organs are enriched in younger genes. Also no doubt as to this being similar to observations made on plants and animals. However, the out of testis hypothesis, originally (Kaessmann 2010 Genome Res), posits that new genes arise in the testis/male organs. For this specifically, the Authors have no supporting data or observations. They should reconsider this part in my opinion.

Minor

l21 - I still think 'evolutionary transcriptome' makes little sense here. A transcriptome is a snapshot of cell states, how can 'evolutionary' be an adjective to it? Evolutionary to me only makes sense as an adjective of nouns like patterns/differences. Maybe say 'evolutionary analysis of transcriptomes' here?

l24 - being more explicit would add clarity, e.g. ...display a non-hourglass like transcriptome that is most conserved...

l119 - for the non-specialist, please state if they have embryos and if embryos were sampled.

l145-147 - fungi don't have an embryogenesis and their hourglass is disputed. Please rephrase.

l209 - I miss a broader conclusion at the end of this section. Can findings in this chapter be contrasted with the previous one (embryogenesis = hourglass, other types of complex development = no hourglass)

l272 - please add which finding is meant to be more broadly tested.

l272 - 284 - including these species is a great way to address my previous points,. However, this section, as written now does not contribute to clarifying whether the hourglass is a property of embryogenesis or not. I still think there is a sampling issue involved. This could be acknowledged and, if the hourglass remain untestable, or dubious, that should be mentioned explicitly. There is no problem I think with ambiguity remaining in the paper; the problem is when clarity is lost.

l312 - in fungi the waist is caused by the upregulation of old genes. Please revise.

l316 - to avoid confusion, instead of complex multicellularity, embryogenesis could be mentioned here.

l327-335 - this chapter is only loosely connected with the storyline and feels like an attempt to mention these previous studies anyway.

Figure 2b, six stages are shown but only 5 pictograms. Maybe worth adding a pictogram to stage 4.5?

Referee #2 (Remarks to the Author):

I think the authors have done a very nice job with their revision- I appreciate the authors' thoughtful responses to my previous comments and the substantial improvements they have made to strengthen their study.

The authors have addressed my main concern about the limited number of species in their analysis by including developmental transcriptomes from two additional kelp species, *Saccorhiza polyschides* and *Laminaria digitata*. This expansion significantly enhances the phylogenetic breadth of their analysis and makes their conclusions much stronger.

I am particularly impressed by how the new datasets reinforce the authors' findings:

1. The multicellular stages in the modular gametophyte stages of the kelp species show similar patterns to those observed in *Ectocarpus*, supporting the generalization of their findings across brown algae clades.
2. The data from *Laminaria digitata*, although missing the very early stages of embryogenesis due to unavoidable limitations of the life cycle (they contain maternal tissue), captures the 'second part' of

the hourglass pattern, showing decreasing transcriptome conservation as development transitions to proliferation stages.

I appreciate how much the authors were able to dive into their datasets to provide additional context for my questions. It's not easy working with non-model systems, and they did a great job addressing my questions, as well as those of the other referees, given these constraints.

Overall, I believe the authors have substantially improved their manuscript. The addition of new species and analyses has strengthened their conclusions and broadened the impact of their work. This study now provides compelling evidence for the generality of the developmental hourglass pattern across complex multicellular eukaryotes, including brown algae.

I recommend this manuscript for publication. It represents a significant contribution to our understanding of the evolution of developmental processes within complex multicellularity.

Referee #2 (Remarks on code availability):

I am not a bioinformatician (at least, not of this sort), so I have not reviewed the code.

Referee #3 (Remarks to the Author):

The authors have carefully responded to all my suggestions. I have no further comments.

Referee #4 (Remarks to the Author):

The revised manuscript is clearly improved. It has benefited from the extensive discussions inspired by the reviewer's comments. The addition of two more species of kelp certainly strengthens the main conclusions despite the technically inaccessible early stages.

The authors addressed my comments regarding the test for hourglass pattern in the multicellular stages of *Ectocarpus*. Although one sex does return a significant hourglass value, the overall conservative interpretation of the data suggesting a lack of an hourglass at those stages seems prudent.

I am still convinced that general readers will be more convinced by a plot of hourglass divergence in gene expression patterns, as opposed to the TAI index, which is a very specific complex metric used primarily by the hourglass community. The authors currently use pure gene expression level divergence (without gene age) only to homologize stages. I think it is a missed opportunity.

The chapter on sex-specific differences in *Fucus* didn't receive much attention. It disrupts the narrative of the paper.

The main conclusion that multicellular life cycle stages in brown algae are associated with constrained transcriptome and that mid-embryogenesis hourglass conservation may be an extreme manifestation of the separation from unicellular stages is an interesting one and could warrant further in-depth discussion. I would like to point the authors to the hypothesis of an egg as an evolutionary novelty by Stuart Newman, more recently rephrased in Kalinka et al. TREE 2012.

Referee #5 (Remarks to the Author):

I have read the authors responses to all of the reviewers' comments. I am satisfied that the authors have addressed my concerns and recommendations and I am pleased to see that they have now included data from other species, as recommended, which increases the robustness of the study. I am also satisfied that the statistical analysis is now sufficient and supports the conclusions.

Referee #5 (Remarks on code availability):

I was able to access and run the code with the information provided.

Author Rebuttals to First Revision:

Referee #1 (Remarks to the Author):

The ms by Lotharukpong et al has improved and I appreciate the detailed responses of the Authors. I think this paper will be an interesting addition to the already sizable literature on the developmental hourglass hypothesis. I have three more substantial suggestions and a few minor ones, that the Authors may find helpful in further improving their ms.

1. A challenge for the reader is that subheadings don't form a coherent storyline, thereby don't help grasping the main message.

Hourglass-shaped transcriptome conservation profiles during *Fucus* embryo development

Distinct evolutionary transcriptomic patterns mark *Fucus* adult tissues and sexes

Multicellularity constrains transcriptome evolution in *Ectocarpus*

Re: We have changed the section headings (also to comply to Nature formatting):

Results section

"Transcriptome evolution in *Fucus* embryogenesis"

"Transcriptome evolution in *Fucus* adults"

"Transcriptome evolution in *Ectocarpus*"

Discussion section

"A transcriptomic hourglass in *Fucus*"

"Young genes in reproductive tissues"

"Multicellularity constrains transcriptome evolution"

Somehow # 2 and 3 don't reflect on the hourglass part of the story and read as if they were addressing very different questions. This lack of coherence is present in the chapters themselves too. I realize this is a complex story, but having a clearer logic would help readers.

Re: We chose to examine adult stages and not only embryogenic stages of the life cycle because we wanted to provide a full picture of the transcriptome evolution landscape in *Fucus*. Although this has not been emphasized in other studies of hourglass, we believe this is interesting because it allows us to investigate transcriptome conservation patterns outside the classical embryogenesis. We have nevertheless proposed alternative section headings to help increase the coherence of the narrative.

2. I think to a large extent clarity is lost on the phrasing. I think it would be important to be very precise about what the hourglass pattern applies to in brown algae. Embryogenesis, complex development, morphologically complex and complex multicellularity are used interchangeably at several places. This causes confusion and is particularly unfortunate given that one of the goals of the paper is to tease apart these concepts and how they relate to an hourglass.

Re: We have revised the wording to make sure it is clear that there is an embryogenetic hourglass in *Fucus* and in algae that go through canonical embryogenesis, whereas in algae that do not go through embryogenesis there are changes in level of transcriptome conservation related to the transitions from unicells to multicellular development. We defined the terms used and checked throughout the use of the different terms and homogenized them (please see the tracked changes version of the ms).

3. 'Out of receptacle' hypothesis. I have reservations about making a comparison with the out of testis hypothesis based on these data. There is no doubt male organs are enriched in younger genes. Also no doubt as to this being similar to observations made on plants and animals. However, the out of testis hypothesis, originally (Kaessmann 2010 Genome Res), posits that new genes arise in the testis/male organs. For this specifically, the Authors have no supporting data or observations. They should reconsider this part in my opinion.

Re: In our understanding, an implication of the original hypothesis of “out of testis” is that young genes are disproportionately expressed in testis. This is exactly what we see in male mature reproductive tissue (the analogous structures to testis) in the brown algae. We have however removed the ‘out of the receptacle’ terms from the manuscript.

Minor

I21 - I still think ‘evolutionary transcriptome’ makes little sense here. A transcriptome is a snapshot of cell states, how can ‘evolutionary’ be an adjective to it? Evolutionary to me only makes sense as an adjective of nouns like patterns/differences. Maybe say ‘evolutionary analysis of transcriptomes’ here?

Re: This has been replaced as suggested throughout the manuscript.

I24 - being more explicit would add clarity, e.g. ...display a non-hourglass like transcriptome that is most conserved...

Re – This sentence has been changed.

I119 - for the non-specialist, please state if they have embryos and if embryos were sampled.

Re: Done (L116) and it is reported in Extended Data Figure 12.

I145-147 - fungi don’t have an embryogenesis and their hourglass is disputed. Please rephrase.

Re: Done (L157)

I209 - I miss a broader conclusion at the end of this section. Can findings in this chapter be contrasted with the previous one (embryogenesis = hourglass, other types of complex development = no hourglass)

Re: Done (L219-221)

I272 - please add which finding is meant to be more broadly tested.

Re: Done (L341-342)

I272 - 284 - including these species is a great way to address my previous points,. However, this section, as written now does not contribute to clarifying whether the hourglass is a property of embryogenesis or not. I still think there is a sampling issue involved. This could be acknowledged and, if the hourglass remain untestable, or dubious, that should be mentioned explicitly. There is no problem I think with ambiguity remaining in the paper; the problem is when clarity is lost.

Re: We have now rephrased terms in this section. With these changes, we address this comment regarding canonical embryogenesis and complex multicellularity. We believe we have shown the presence of a hourglass during the embryogenesis of complex brown algae, and we have explained in the text the difficulty in sampling more stages for additional non-model algae.

I312 - in fungi the waist is caused by the upregulation of old genes. Please revise.

Re: We have removed the citation to fungi (L368)

I316 - to avoid confusion, instead of complex multicellularity, embryogenesis could be mentioned here.

Re: This has been done (L372)

I327-335 - this chapter is only loosely connected with the storyline and feels like an attempt to mention these previous studies anyway.

Re: We think this sentence make sense because they allow to connect the transcriptomic hourglass to the morphological changes during embryogenesis in complex brown algae, so we would prefer to keep it.

Figure 2b, six stages are shown but only 5 pictograms. Maybe worth adding a pictogram to stage 4.5?

Re: This has been done

Referee #2 (Remarks to the Author):

I think the authors have done a very nice job with their revision- I appreciate the authors’ thoughtful

responses to my previous comments and the substantial improvements they have made to strengthen their study.

The authors have addressed my main concern about the limited number of species in their analysis by including developmental transcriptomes from two additional kelp species, *Saccorhiza polyschides* and *Laminaria digitata*. This expansion significantly enhances the phylogenetic breadth of their analysis and makes their conclusions much stronger.

I am particularly impressed by how the new datasets reinforce the authors' findings:

1. The multicellular stages in the modular gametophyte stages of the kelp species show similar patterns to those observed in *Ectocarpus*, supporting the generalization of their findings across brown algae clades.

2. The data from *Laminaria digitata*, although missing the very early stages of embryogenesis due to unavoidable limitations of the life cycle (they contain maternal tissue), captures the 'second part' of the hourglass pattern, showing decreasing transcriptome conservation as development transitions to proliferation stages.

I appreciate how much the authors were able to dive into their datasets to provide additional context for my questions. It's not easy working with non-model systems, and they did a great job addressing my questions, as well as those of the other referees, given these constraints.

Overall, I believe the authors have substantially improved their manuscript. The addition of new species and analyses has strengthened their conclusions and broadened the impact of their work. This study now provides compelling evidence for the generality of the developmental hourglass pattern across complex multicellular eukaryotes, including brown algae.

I recommend this manuscript for publication. It represents a significant contribution to our understanding of the evolution of developmental processes within complex multicellularity.

Referee #2 (Remarks on code availability):

I am not a bioinformatician (at least, not of this sort), so I have not reviewed the code.

Referee #3 (Remarks to the Author):

The authors have carefully responded to all my suggestions. I have no further comments.

Referee #4 (Remarks to the Author):

The revised manuscript is clearly improved. It has benefited from the extensive discussions inspired by the reviewer's comments. The addition of two more species of kelp certainly strengthens the main conclusions despite the technically inaccessible early stages.

The authors addressed my comments regarding the test for hourglass pattern in the multicellular stages of *Ectocarpus*. Although one sex does return a significant hourglass value, the overall conservative interpretation of the data suggesting a lack of an hourglass at those stages seems prudent.

I am still convinced that general readers will be more convinced by a plot of hourglass divergence in gene expression patterns, as opposed to the TAI index, which is a very specific complex metric used primarily

by the hourglass community. The authors currently use pure gene expression level divergence (without gene age) only to homologize stages. I think it is a missed opportunity.

Re: We would prefer to leave the TAI in main figure and the hourglass divergence in gene expression patterns as a supplemental figure – we had already added the figures for *F. serratus* vs *F. distichus*, as well as *Ectocarpus* vs *Fucus* in terms of transcriptome divergence (please see Extended Data Figure 4 and Extended Data Figure 8). We added in L156, further interpretation of Extended Data Figure 4, which indicates the relative divergence in early and late stages at the pure gene expression level.

The chapter on sex-specific differences in *Fucus* didn't receive much attention. It disrupts the narrative of the paper.

Re: We hope that with the clarifications, changes in subheadings and simplification of terms the narrative is smooth and the section does not disrupt the narrative. We would prefer to keep this small section.

The main conclusion that multicellular life cycle stages in brown algae are associated with constrained transcriptome and that mid-embryogenesis hourglass conservation may be an extreme manifestation of the separation from unicellular stages is an interesting one and could warrant further in-depth discussion. I would like to point the authors to the hypothesis of an egg as an evolutionary novelty by Stuart Newman, more recently rephrased in Kalinka et al. TREE 2012.

Re: We appreciate the reviewer's point and we have now added the suggested reference with regards to the adaptive ideas for the *Ectocarpus* unicells (L479). However, an in-depth discussion would have to be lengthy, and we would prefer not to over-speculate on this matter.

Referee #5 (Remarks to the Author):

I have read the authors responses to all of the reviewers' comments. I am satisfied that the authors have addressed my concerns and recommendations and I am pleased to see that they have now included data from other species, as recommended, which increases the robustness of the study. I am also satisfied that the statistical analysis is now sufficient and supports the conclusions.

Referee #5 (Remarks on code availability):

I was able to access and run the code with the information provided.